# DocMIA: Document-Level Membership Inference Attacks against DocVQA Models

**Khanh Nguyen**[1]   **Raouf Kerkouche**[2]*   **Mario Fritz**[2]   **Dimosthenis Karatzas**[1]

[1]Computer Vision Center, Universitat Autònoma de Barcelona
[2]CISPA Helmholtz Center for Information Security
{knguyen,dimos}@cvc.uab.es
{raouf.kerkouche,fritz}@cispa.de

## Abstract

Document Visual Question Answering (DocVQA) has introduced a new paradigm for end-to-end document understanding, and quickly became one of the standard benchmarks for multimodal LLMs. Automating document processing workflows, driven by DocVQA models, presents significant potential for many business sectors. However, documents tend to contain highly sensitive information, raising concerns about privacy risks associated with training such DocVQA models. One significant privacy vulnerability, exploited by the membership inference attack, is the possibility for an adversary to determine if a particular record was part of the model's training data. In this paper, we introduce two novel membership inference attacks tailored specifically to DocVQA models. These attacks are designed for two different adversarial scenarios: a white-box setting, where the attacker has full access to the model architecture and parameters, and a black-box setting, where only the model's outputs are available. Notably, our attacks assume the adversary lacks access to auxiliary datasets, which is more realistic in practice but also more challenging. Our unsupervised methods outperform existing state-of-the-art membership inference attacks across a variety of DocVQA models and datasets, demonstrating their effectiveness and highlighting the privacy risks in this domain.

## 1 Introduction

Automated document processing fuels a significant number of operations daily, ranging from fintech and insurance procedures to interactions with public administration and personal record keeping. Up until a few years ago, document processing services relied on template-based information extraction models, which were created ad-hoc for each client. Although these approaches allowed for good control of client data and could be extended to new documents with a few examples, they were limited in scalability and difficult to maintain. Consequently, the introduction of Document Visual Question Answering (DocVQA) (Mathew et al., 2020) in 2019 has resulted in a paradigm shift in document processing services, enabling end-to-end generic solutions to be applied in this domain. DocVQA leverages multi-modal large language models to streamline business workflows and provide clients with novel ways to interact with the document processing pipeline.

However, as cloud-based DocVQA solutions become more prevalent, significant privacy risks emerge, particularly concerning the potential leakage of sensitive information through model vulnerabilities. Indeed, during the training of a DocVQA model, *each document can have several associated question-answer pairs*, with each pair considered a unique data point. As a result, a single document can appear multiple times, which significantly raises the risks associated with privacy vulnerabilities. This repeated exposure enhances the likelihood of the model memorizing specific details, thereby increasing the potential for data leakage during privacy attacks. Furthermore, scanned document images often have high resolutions necessary for posterior analysis, but need to be rescaled for processing by image encoders, potentially rendering content unreadable. To mitigate this issue, many DocVQA models (Huang et al., 2022; Tang et al., 2023) utilize a dual representation

---

*The corresponding author.

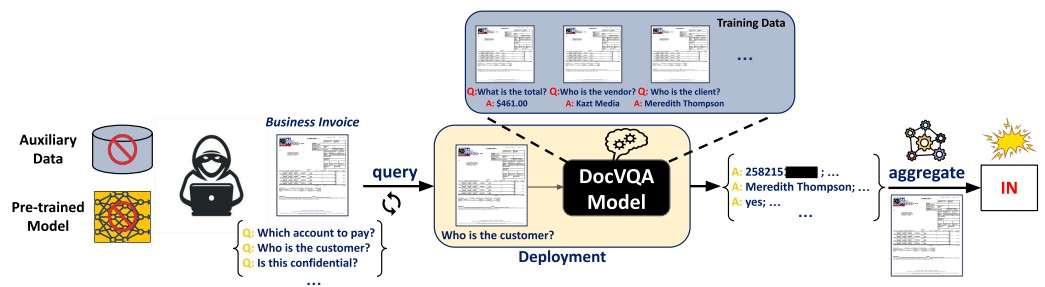

Figure 1: **The General Scheme of Document-level Membership Inference Attacks.** *Training:* A DocVQA model is trained on a dataset of documents, each associated with **multiple questions/answers**. *Deployment:* An adversary exploits this structure by querying the model with questions related to a target document. By aggregating the model's responses, the adversary can infer the membership of the document in the training set.

of the document, comprising both a reduced-scale image and OCR-recognized text. This approach introduces further challenges, as sensitive information may leak through multiple modalities.

Membership inference attacks (MIAs) are among the most prominent techniques for assessing privacy vulnerabilities in machine learning models. These attacks enable an adversary to determine whether a specific data point is included in the training dataset. However, there is limited research on membership inference risks in the context of multi-modal models. Among the few studies, Ko et al. (2023); Hu et al. (2022b) utilize powerful pre-trained models on large datasets to construct an aligned embedding space for the two modalities—image and text—allowing for the inference of membership information. Unfortunately, the reliance on these pre-trained models poses challenges for document-based tasks, particularly in DocVQA scenarios, where an alignment model capable of aligning the (document, question) as input and the answer as output is currently unavailable. Recently, Tito et al. (2024) introduced a provider-level MIAs against DocVQA models aimed at determining whether a *provider (group)* that may supply multiple invoice documents is part of the training set. In contrast, our research focuses on membership information at a finer granularity, specifically targeting the inference of whether a *single document* is included in the training dataset. Current MIA solutions that exploit standard features such as output logits, probabilities, or loss are difficult to adapt to the DocVQA context, where outputs are generated in an auto-regressive manner. Additionally, legal constraints surrounding copyright and private information complicate centralized model training, making it challenging to create auxiliary datasets that capture the variability and richness of real-world data. As a result, shadow training of proxy models becomes infeasible.

In this work, we take a structured approach to privacy testing for DocVQA models. We design a novel Document-level Membership Inference Attack (DocMIA) that deals with the multiple occurrences of the same document in the training set, as demonstrated in Figure 1. To address the challenge of extracting typical metrics (e.g. logit-based) from auto-regressive outputs, we propose a new method based on model optimisation for individual samples that generates discriminative features for DocMIA. We design attacks both for white-box and black-box settings without requiring auxiliary datasets. In the black-box setting, we propose an alternative knowledge transfer mechanism from the attacked model to a proxy. Evaluating our attacks on three multi-modal DocVQA models and two datasets, we achieve state-of-the-art performance against multiple baselines.

To summarize, we make the following contributions:

1. We present DocMIA, the first Document-level Membership Inference Attacks specifically targeting multi-modal models for DocVQA.

2. We introduce two novel auxiliary data-free attacks for both white-box and black-box settings, leveraging novel discriminative metrics for DocMIA.

3. We explore three distinct approaches to quantify these metrics: vanilla layer fine-tuning (FL), fine-tuning layer with LoRA (Hu et al., 2021) (FLLoRA), and image gradients (IG).

4. Our attacks[1], evaluated on two DocVQA datasets across three different models, outperform existing state-of-the-art membership inference attacks as well as baseline attacks.

---

[1]Code is available at `https://github.com/khanhnguyen21006/mia_docvqa`

## 2 RELATED WORK

**Membership Inference Attack.** Membership inference attacks have been extensively explored in various applications to highlight privacy vulnerabilities in deep neural networks or to audit model privacy (Shokri et al., 2017). These attacks are categorized into two types: white-box and black-box settings. In white-box settings, the adversary has full access to the target model's internal parameters and computations (Carlini et al., 2022; Yeom et al., 2018; Nasr et al., 2019; Rezaei & Liu, 2021; Sablayrolles et al., 2019; Li & Zhang, 2021), enabling the use of informative features like loss values, logits, and gradient norms. Conversely, in black-box settings, the adversary is limited to the model's outputs, such as predicted labels or confidence scores (Choquette-Choo et al., 2021; Shokri et al., 2017; Salem et al., 2018; Sablayrolles et al., 2019; Song & Mittal, 2021; Hui et al., 2021). The literature indicates that white-box attacks tend to be more effective due to the availability of richer features (Song et al., 2019; Nasr et al., 2019). In this paper, we propose tailored attacks for both settings, considering a more challenging scenario where the adversary lacks an auxiliary dataset –which is used to train shadow models that mimic the behavior of the target model and are subsequently exploited to enhance attack performance– and is restricted to a limited number of queries. Regarding gradient-based membership inference attacks, research on using gradients as features has been limited. Nasr et al. (2019) leveraged the $L2$-norm of gradients with respect to model weights for membership inference. Rezaei & Liu (2021) suggested using the distance to the decision boundary as a metric but found it ineffective for this purpose. In contrast, we introduce novel strategies called FL, FLLoRA, and IG, demonstrating that the $L2$-norm of the cumulative gradient—computed using these methods—provides a robust signal for membership inference. While Maini et al. (2021) and Li & Zhang (2021) also explored distance metrics, but from input points for membership inference in image classification tasks, their approaches lack scalability and applicability in our context, which involves larger-scale models with a wider vocabulary of tokens.

**Membership Inference Attack Against Multi-modal Models.** Research works into the privacy vulnerabilities of multi-modal models is still in its early stages. Recently, Tito et al. (2024); Pinto et al. (2024) proposed reconstruction attacks that exploit DocVQA model memorization to recover hidden values in documents. They black out specific target values in documents and query the model with questions about the modified documents. Since the model memorizes training data, it often reconstructs the hidden target values. Tito et al. (2024) also introduced a membership attack against DocVQA models to infer whether a document provider, with multiple documents, is included in the training dataset. However, as far as we know, no research has yet explored membership inference attacks at document-level granularity. Additionally, Ko et al. (2023); Hu et al. (2022b) leverage powerful *pre-trained models* on large datasets to create an aligned embedding space for the two modalities to infer membership. Unfortunately, the reliance on these pre-trained models introduces difficulties for document-based tasks, especially DocVQA, where an appropriate alignment model for aligning (document, question) inputs to corresponding answers is not yet available. Furthermore, the success of both attacks hinges on the availability of *auxiliary datasets* leveraged by the adversary, which are key to executing the attack effectively. In this paper, we present two membership inference attacks specifically tailored to tackle the unique characteristics of DocVQA models.

## 3 BACKGROUND

### 3.1 DOCUMENT-BASED VISUAL QUESTION ANSWERING

DocVQA is a multi-modal task where natural language questions are posed based on the content of document images. Notably, it establishes a unified query-response framework applicable across various document understanding tasks, such as document classification and information extraction.

Formally, the DocVQA task is defined as follows: given a question-answer pair $(q, a)$ related to a document image $x$, the method $\mathcal{F}$ must generate an answer $\hat{a} = \mathcal{F}(x, q)$ such that $\hat{a}$ closely matches the correct answer $a$. More concretely, given $D_t = \{(x_i, q_i, a_i)\}_{i=1}^{N_t}$ as a set of valid training examples, a model $\mathcal{F}$, parameterized by $\theta$, is trained to maximize the conditional log-likelihood:

$$\mathcal{L}(\theta) = -\log p_\theta(a_i | x_i, q_i) \tag{1}$$

Standard metrics for DocVQA include Accuracy (ACC) and Normalized Levenshtein Similarity (NLS) (Biten et al., 2019), which measure the similarity between the predicted and correct answer:

$$\text{ACC} = \mathbf{1}_{\hat{a}=a}; \quad \text{NLS} = \begin{cases} 1 - \text{NL}(\hat{a}, a) & \text{if NL}(\hat{a}, a) < 0.5, \\ 0 & \text{if NL} \geq 0.5 \end{cases} \quad (2)$$

where $\text{NL}(\cdot, \cdot)$ denotes the normalized Levenshtein distance.

In the following sections, for clarity, we often omit the data example index $i$ from the notation.

## 3.2 DOCUMENT-LEVEL MEMBERSHIP INFERENCE ATTACK

Membership inference attacks (Shokri et al., 2017) exploit privacy vulnerabilities to determine if a specific data point was included in the training set of a machine learning model. We extend this definition to the Document-level MIA, which is particularly suited in the DocVQA context.

Given access to a trained DocVQA model $\mathcal{F}$ and a document $x$ drawn from its data distribution $\mathcal{D}$, along with a set of question-answer pairs $Q = \{(q_i, a_i)\}_{i=1}^{M}$ related to the information in the document, an adversary $\mathcal{A}$ designs a decision rule $f_{\mathcal{A}}(x, Q; \mathcal{F})$ to classify the membership status of $x$, aiming for $f_{\mathcal{A}}(x, Q; \mathcal{F}) = 1$ if $x$ is a member of the training set, otherwise a non-member. It is important to note that the adversary is focused solely on *the membership of the document $x$*, rather than the entire DocVQA *data point* $(x, q, a)$, which is typically the target of prior MI attacks. Moreover, since a single document is associated with many question-answer pairs, this allows the adversary to query the same document using multiple questions for various pieces of information.

# 4 DOCMIA AGAINST DOCVQA MODELS

In Section 4.1, we elaborate on the threat model relevant to DocMIA on two scenarios: *white-box* and *black-box* access. We first explain our intuition behind our optimization-based attacks in the white-box setting (Section 4.2), then adapt this approach to our black-box attacks (Section 4.3).

## 4.1 THREAT MODEL

DocMIA can be either a useful or harmful tool in various real-world scenarios. On the positive side, DocMIA can act as a privacy auditing tool. For instance, in legal document processing, law firms may use these attacks to evaluate whether proprietary or confidential documents, such as contracts or court filings, were included in model training, thereby identifying potential privacy risks. Conversely, DocMIA can be maliciously leveraged. As an example, a business competitor could exploit these attacks on an invoice-processing system to infer the presence of specific invoices in the training data, exposing confidential business relationships and leading to risks such as supplier poaching.

In both scenarios, we assume that the adversary aims to infer membership information for a set of documents, determining whether each document is included in the training dataset. These documents may or may not be part of the target model's training data. Crucially, we further assume *the adversary lacks access to an auxiliary data $D_{\text{aux}}$* that reflects the characteristics of these documents. This assumption is realistic, as obtaining real-world documents at scale is often prohibitively difficult due to their confidential nature and regulatory restrictions. Consequently, this negates the application of MI attack techniques that require training shadow models (Shokri et al., 2017; Carlini et al., 2022). Even if auxiliary documents were available, training numerous shadow document-based models—typically designed with a large number of parameters—would be prohibitively expensive.

Based on the previous examples, we refer to the owner of the document model as *the trainer* and the law firms or competitors as *the adversary*. Given the document distribution $\mathcal{D}$, the trainer trains a document-based model $\mathcal{F}_t$ with private access to $D_t \sim \mathcal{D}$, following a training algorithm $\mathcal{T}$, that defines the model architecture, optimization process, and related details. The adversary owns the set of sensitive documents $D_{\text{test}} \sim \mathcal{D}$, where $D_t \cap D_{\text{test}} \neq \emptyset$, $|D_{\text{test}}| = N_{\text{test}}$; but does not know which documents are in $D_t$. Given a document $x \in D_{\text{test}}$ with a set of related queries $Q = \{(q_i, a_i)\}_{i=1}^{M}$, the adversary's goal is to determine whether $x \in D_t$ or $x \notin D_t$. We formulate two attack settings, which specify the adversarial knowledge about the model $\mathcal{F}_t$ and its data distribution $\mathcal{D}$:

**White-box Setting**. In this scenario, the adversary has full access to the internal workings of the target model, including the model's architecture, weights, gradients from any further training and other internal details. However, the adversary does not have access to the training algorithm $\mathcal{T}$.

**Black-box Setting**. Here, the adversary can only interact with the target model through an API, which only returns a prediction $\hat{a}$ for each question $q$ on $x$. In addition, the adversary is constrained by a limited number of queries. As in the white-box setting, the adversary has no information about $\mathcal{T}$. This setting reflects the most challenging case (Nasr et al., 2019; Song et al., 2019).

We assume the adversary has full knowledge of the DocVQA task to train the model, including the training objective, document type and exact training questions. This assumption is reasonable, as task-level information such as document type, is often publicly available to guide users, making it accessible to adversaries. The assumption of the exact question knowledge is also plausible, as an adversary can *approximate* questions based on the document type. Further discussion of this assumption and experiments in the setting without exact questions are provided in Appendix F.2.

## 4.2 WHITE-BOX DOCMIA

In the white-box setting, where the adversary has access to the trained model, shadow training is impractical due to the lack of auxiliary data and high computational cost. To this end, our strategy is to develop unsupervised *metric-based* attacks (Hu et al., 2022a). For each document, we extract a set of features from individual question-answer pairs and aggregate them across all pairs. We then cluster the resulting feature vectors to distinguish member from non-member documents. A key challenge is to design discriminative features, as standard metrics (e.g., logit and loss) may be ineffective in this setting (Section 6). To address this, we propose new features that enhance the informativeness of our membership inference vectors.

### 4.2.1 OPTIMIZATION-BASED DISCRIMINATIVE FEATURES

In this section, we introduce two novel discriminative membership features derived from an optimization process for our attacks against DocVQA models.

**Intuition.** Since DocVQA models are typically trained on multiple question-answer pairs per document, the model parameters likely converge to *minimize the average distance to the ground-truth answers* after training. As a result, fine-tuning the model on one question-answer pair through an iterative process is necessary to extract more reliable membership signals. More importantly, this optimization on training documents may converge faster than to non-training documents, due to the lower generalization error. Figure 2 illustrates our reasoning.

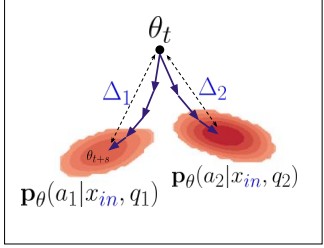

$x$ is **in** of the training set

We provide a formal definition of the *distance* feature.

**Definition 4.1 (Optimization-based Distance Feature).** Given a model $\mathcal{F}$ parameterized by $\theta$, let the model be initialized with $\theta_0$. After undergoing a gradient-based optimization process $\mathcal{O}$, the parameters converge to $\theta^*$ according to a specified training objective $\mathcal{L}$. The *distance* feature is then defined as the $L2$-norm of the change in parameters:

$$\Delta(\theta_0, \theta^*) = ||\theta_0 - \theta^*||_2 \quad (3)$$

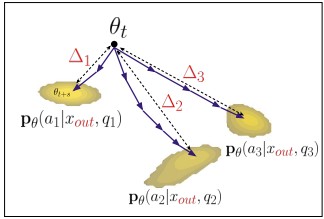

$x$ is **out** of the training set

This feature measures the difference between the initial parameters $\theta_0$ and the converged parameters $\theta^*$, as an approximation of the optimization trajectory toward the optimal solution.

Specifically, we fine-tune the target DocVQA model on an individual document/question-answer pair and compute the *distance* required to reach the *optimal* answer. A small average distance indicates the document is likely part of the training set, while a larger distance suggests a non-training document. In addition, the number of optimization steps serves as an orthogonal feature that reflects the efficiency of the optimization process. With an optimal learning

Figure 2: **Visualization of our fine-tuning strategy** in the parameters space. Each contour plot represents the optimization landscape w.r.t each pair $(a_i, q_i)$ from document $x$. In general, the average $\Delta$ computed on a member document $x_{\text{in}}$ is smaller than non-member document $x_{\text{out}}$.

rate and a good initialization provided from the target model, optimization for training documents typically converges in fewer steps compared to non-training documents. Consequently, we include both the *distance* and the number of optimization steps in our feature set for white-box attacks.

---

**Algorithm 1** DocMIA Assignment

---

1: **Input:** model $\mathcal{F}_{\theta_t}$, document $x \in D_{\text{test}}$, question-answer pairs $\{(q_i, a_i)\}_{i=1}^M$, utility $\mathcal{U}$, aggregation $\Phi$.
2: **Hyperparameters:** optimizer OPT, optimization steps $S$, learning rate $\alpha$, threshold $\tau$.
3: **for** $i = 1$ **to** $M$ **do**
4:     Set $\theta$.requires_grad = True     // Change $\theta$ to: $\theta_L$ or LoRA($\theta_L$) or $x$ and Freeze $\theta$.
5:     Initialize: $s_i = 0, u_i = \{\}; l_i \leftarrow 0, \theta_0 \leftarrow \theta_t$
6:     **while** $s_i < S$ **do**
7:         $u_i \leftarrow u_i \cup \mathcal{U}(\mathcal{F}_\theta(x, q_i), a_i)$
8:         **if** $(\mathcal{L}(\theta) - l) < \tau$ **then** break;     // Early stopping
9:         **end if**
10:        $\theta \leftarrow \text{OPT}(\alpha, \nabla_\theta(\mathcal{L}(\theta))$
11:        $l_i \leftarrow \mathcal{L}(\theta), s_i \leftarrow s_i + 1$
12:     **end while**
13:     $\Delta_i \leftarrow \|\theta_0 - \theta\|_2$     // Compute distance metric
14: **end for**
15: $\Delta_M = \Phi(\Delta_{i=1,\ldots,M}); s_M = \Phi(s_{i=1,\ldots,M}); u_M = \Phi(u_{i=1,\ldots,M})$     // Aggregating over $M$ questions
16: **Output:** $F_x = [\Delta_M, s_M, u_M]$     // Assign membership feature vector

---

### 4.2.2 METHODOLOGY

We now formally present our attack strategy, assuming white-box access to the target model $\mathcal{F}_t$. For any document $x \in D_{\text{test}}$ and a set of question-answer pairs $Q$, the goal is to assign a features descriptor $F_x$. This is achieved by first extracting a set of features through the optimization process $\mathcal{O}$ on a single question-answer pair. These features are aggregated across multiple questions then concatenated to construct $F_x$. Repeating this process over $D_{\text{test}}$, we apply an unsupervised clustering algorithm to differentiate member documents from non-members based on their features descriptors.

Following our intuition, for each question-answer pair $(q, a)$, we fine-tune the target model parameter $\theta_t$ via gradient descent to maximize $p_\theta(a|x, q)$ (Eq. 1). Starting from $\theta_t$, optimization proceeds with learning rate $\alpha$, querying the model at each step $s$ using $q$ and evaluating prediction $\hat{a}$ via utility function $\mathcal{U}$, either ACC or NLS (Eq. 2). The process stops when no further improvements is observed, governed by threshold $\tau$ or max $S$ steps. We then evaluate the distance $\Delta$ (Eq. 3), record the number of steps $s$, and aggregate the utility evolution to obtain the overall DocVQA score $u$. These features serve as membership signals for the current $(q, a)$ pair in relation to the target document $x$.

Since each document is associated with a varying number of question-answer pairs $M$, we aggregate features across all pairs using $\Phi$, optionally employing multiple aggregation functions to enrich the feature set. The aggregated features are then normalized onto a consistent scale and concatenated to form the document descriptor $F_x$. The assignment algorithm is outlined in Algorithm 1. Finally, we cluster descriptors from $D_{\text{test}}$, identifying the cluster with the larger $\Delta$ as non-member documents.

### 4.2.3 IMPROVING EFFICIENCY

Fine-tuning $\mathcal{F}_t$ on a *single* $(q, a)$ enables membership inference but is computationally expensive due to model size and data pre-processing overhead. To improve the attack efficiency, we introduce three variants of the method, as illustrated in Figure 3:

**Optimize One Layer (FL).** Instead of updating all parameters, we optimize a single layer, assuming its gradients provide sufficient signal for membership classification. In this variant, we select one specific layer $L$ to optimize, keeping the other parameters fixed. We ablate the choice of $L$ for this method in Appendix C. In addition, we propose **FLLoRA**, a variant using LoRA (Hu et al., 2021), where the LoRA parameters are initialized with Kaiming initialization (He et al., 2015). From Algorithm 1, we replace $\theta$ to $\theta_L$ or the LoRA parameters of $L$, denoted as LoRA($\theta_L$), respectively.

**Optimize the Document Image (IG).** This variant shifts the focus to the input space by directly optimizes the pixel values of the document image $x$. The intuition remains the same: training documents require less self-tuning, enabling the model to converge faster than non-trainings. However, this assumes the target model allows differentiation of the document image through its architecture. Accordingly, we replace $\theta$ with $x$ from Algorithm 1 while freezing the target model parameters $\theta$.

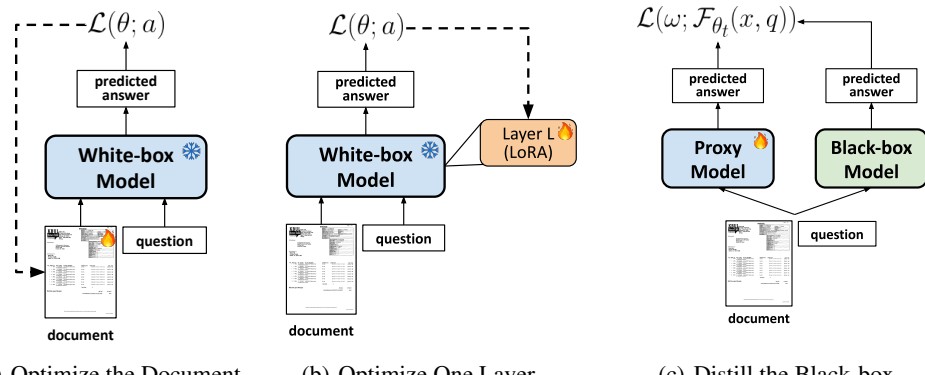

(a) Optimize the Document     (b) Optimize One Layer     (c) Distill the Black-box

Figure 3: **Variants of our proposed DocMIA.** Left: (a) (b) illustrate three white-box attack strategies: optimizing either the Document Image or a single Layer (LoRA). Dashed arrows indicate back-propagated gradients. Right: We distill the black-box model into a proxy, which is then attacked using the white-box strategies.

These variants reduce computational costs while maintaining attack performance, providing more practical options when the size of $D_{\text{test}}$ increases.

### 4.3 BLACK-BOX DOCMIA

In the black-box setting, the attack model's access is restricted to $D_{\text{test}}$ and the predicted answers. To address this, we propose a distillation-based attack that transfers knowledge about the private data $D_t$ from the black-box model $\mathcal{F}_t$ to a proxy model $\mathcal{F}_p$, parameterized by $\omega$. With full control over $\mathcal{F}_p$, we can apply white-box attack strategies without modification.

First, the black-box model generates labels for each question in $D_{\text{test}}$, creating a query dataset $D_{\text{query}} = \{(x_i, Q_i)\}_{i=1}^{N_{\text{test}}}$ where $Q_i = \{(q_j, \mathcal{F}_{\theta_t}(x_i, q_j))\}_{j=1}^{M}$. The proxy model $\mathcal{F}_p$ is then trained on this dataset to maximize the likelihood of the predicted answer $p_\omega(\mathcal{F}_{\theta_t}(x_i, q_j)|x_i, q_j)$. In essence, the goal is to replicate the black-box model's predictions and transfer its output structure to the proxy model, thereby transferring the embedded membership features and validating the white-box attack assumptions. Figure 3(c) illustrates the scheme of our proposed attack.

Since our focus is on the document domain, we initialize $\mathcal{F}_p$ using a publicly available checkpoint $\omega_{\text{pt}}$, pre-trained on an unlabeled document dataset $D_{\text{pt}}$, which is *inaccessible* and assumed to be *disjoint* from the private dataset $D_t$. This initialization equips the proxy model with document understanding capabilities without prior knowledge of $D_t$. As a result, it enables the proxy $\mathcal{F}_p$ to better mimic the prediction behavior and internal dynamics of the black-box $\mathcal{F}_t$ after fine-tuning.

It is important to note that the adversary lacks information of the black-box training algorithm $\mathcal{T}$, meaning there is no advantage regarding model architecture or training details when constructing the proxy model. Thus, the choice of the proxy model, optimizer, learning rate, etc., is independent of the target model. However, as empirically shown in Section 6.1, while using a proxy model with the same architecture as the black-box model can offer benefits, our attack strategies remain effective even with different architectures. This demonstrates the robustness of our approach, which does not rely on specific model classes or requiring detailed knowledge of the black-box model.

## 5 EXPERIMENTAL SETUP

### 5.1 TARGET DATASET AND MODEL

**Target Dataset**. We study two established DocVQA datasets in the literature for our analysis: **DocVQA (DVQA)** (Mathew et al., 2021) and **PFL-DocVQA (PFL)** (Tito et al., 2024). Both datasets are designed for extractive DocVQA task, where the answer is explicitly found within the document image. Each document in these datasets is accompanied by varying number of questions.

**Target Model**. We consider three state-of-the-art models which are designed for document understanding tasks: (1) **Visual T5 (VT5)** (Tito et al., 2024) (250M parameters) follows the traditional design by utilizing OCR module to facilitate the reasoning process. It leverages the T5 model pre-

| | Model | SCORE-TA | | SCORE-UA | | SCORE-UA$_{all}$ | | LOSS-TA | | GRADIENT-UA | | SCORELOSS-UA$_{all}$ | |
|---|---|---|---|---|---|---|---|---|---|---|---|---|---|
| | | ACC | F1 | ACC | F1 | ACC | F1 | ACC | F1 | ACC | F1 | ACC | F1 |
| PFL | VT5 | **62.33** | **64.13** | $61.00_{0.0}$ | $68.80_{0.0}$ | $60.67_{0.0}$ | $60.67_{0.0}$ | 57.83 | 62.81 | $60.67_{0.0}$ | $60.67_{0.0}$ | $\mathbf{60.67_{0.0}}$ | $\mathbf{60.67_{0.0}}$ |
| | Donut | **73.33** | **75.14** | $68.33_{0.0}$ | $75.77_{0.0}$ | $71.17_{0.67}$ | $71.33_{3.32}$ | **73.67** | **78.99** | $70.67_{0.0}$ | $69.55_{0.0}$ | $70.83_{0.0}$ | $69.67_{0.0}$ |
| DVQA | VT5 | **75.67** | **75.75** | $72.17_{0.0}$ | $76.18_{0.0}$ | $75.17_{0.0}$ | $75.13_{0.0}$ | 73.67 | 77.99 | $71.17_{0.0}$ | $67.54_{0.0}$ | $75.50_{0.0}$ | $76.02_{0.0}$ |
| | Donut | 79.67 | 79.53 | $75.97_{0.07}$ | $79.57_{0.07}$ | $\mathbf{80.50_{0.0}}$ | $\mathbf{81.10_{0.0}}$ | 51.83 | 53.46 | $77.17_{0.0}$ | $75.92_{0.0}$ | $\mathbf{80.50_{0.0}}$ | $\mathbf{81.10_{0.0}}$ |
| | Pix2Struct-B | 67.33 | 67.97 | $\mathbf{68.17_{0.0}}$ | $\mathbf{71.36_{0.0}}$ | $69.13_{0.07}$ | $67.67_{0.09}$ | 59.33 | 64.63 | $66.0_{0.0}$ | $68.32_{0.0}$ | $69.00_{0.0}$ | $67.48_{0.0}$ |

Table 1: **Results from Baseline Attacks.** Gray color indicate attacks conducted in the black-box setting. All results are reported based on five random seeds for KMEANS. The methods with the best *average* performance across the two metrics are highlighted in **bold**. See Appendix E for the complete results.

trained on the C4 corpus (Raffel et al., 2020), along with a ViT backbone pre-trained on document data (Li et al., 2022). (2) **Donut** (Kim et al., 2022) (201M parameters) is one of the first end-to-end DocVQA models capable of achieving competitive performance without relying on OCR. It is pre-trained on a large collection of private synthetic documents. (3) **Pix2Struct** (Lee et al., 2023) is another OCR-free document model with two versions: Base (282M) and Large (1.3B parameters). This model is pre-trained to perform semantic parsing on an 80M subset of the C4 corpus.

For the PFL-DocVQA dataset, we consider two targets: VT5, using the public checkpoint provided by the authors[2], and Donut, which we successfully trained to achieve strong performance following the training procedure from the authors. For the DocVQA dataset, we attack four targets: VT5, Donut, and Pix2Struct (Base and Large), all with publicly available checkpoints from Hugging-Face[3] (Wolf et al., 2020). In the black-box setting, we use VT5 and Donut as proxy models. To train the proxy models on the query set $D_{query}$, we initialize them with their public *pre-trained* checkpoints—the same checkpoints used to fine-tune the target models on the respective target datasets, as outlined in the original papers. For more details on datasets and models, see Appendix A and D.

## 5.2 IMPLEMENTATION

We tune the hyperparameters in the optimization process to ensure our attacks are effective against each target model in the white-box setting, then apply the best set to black-box attacks. Assuming no access to the training algorithm $\mathcal{T}$, we use Adam (Kingma, 2014) as the optimizer OPT across all attack experiments. We explore the impact of learning rate $\alpha$, the selected layer $L$, and we carefully tune the values of threshold $\tau$ in the ablation study (Appendix C). Subsequently, the optimal set of hyperparameters for each model is then applied in all black-box experiments. For aggregation $\Phi$, we consider 4 aggregation functions {AVG; MIN; MAX; MED} for each feature, denoted as $\Phi_{all}$. Throughout our experiments, we employ KMEANS as the clustering algorithm. See Appendix D for more implementation details.

## 5.3 EVALUATION METRIC

From the official splits of each target dataset, we sample 300 member documents from the training set and 300 non-member documents from the test set, yielding $N_{test} = 600$ test documents. We evaluate attack performance using *Balanced Accuracy* and *F1 score* in the balanced setting, as in prior works (Salem et al., 2018; Watson et al., 2022; Ye et al., 2022). In addition, we report *True Positive Rate (TPR) at 1% and 3% False Positive Rate (FPR)*, as in recent MIA literature (Carlini et al., 2022). For unsupervised attacks using KMEANS, the membership score of each document is computed as the Euclidean distance between its feature vector and the member cluster centroid.

## 5.4 BASELINE

In the **black-box** setting, we evaluate three MI attacks that only requires the predicted answer to determine membership: Score-Threshold Attack (SCORE-TA), Unsupervised Score-based Attack with AVG (SCORE-UA) (Tito et al., 2024) and $\Phi_{all}$ (SCORE-UA$_{all}$).

For the **white-box** setting, where loss or gradient is accessible, we evaluate three baselines: Loss-Threshold Attack (LOSS-TA) (Yeom et al., 2018), Unsupervised Score+Loss Attack (SCORELOSS-UA$_{all}$) and Unsupervised Gradient Attack (GRADIENT-UA)(Nasr et al., 2019). In

---

[2] https://benchmarks.elsa-ai.eu/?ch=2
[3] https://huggingface.co/models

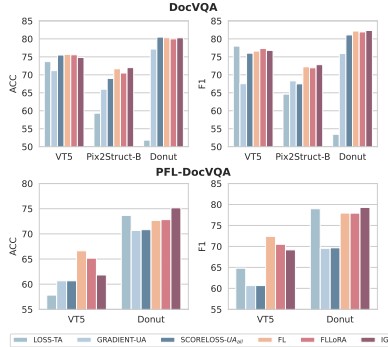

| | DVQA | | | PFL | |
|---|---|---|---|---|---|
| | VT5 | Donut | Pix2Struct-B | VT5 | Donut |
| LOSS-TA | **14.00** | 7.67 | 5.33 | 3.00 | **14.67** |
| GRADIENT-UA | 9.33 | 6.00 | 5.00 | 3.00 | 8.33 |
| SCORELOSS-UA$_{all}$ | 4.67 | 8.67 | 6.67 | 4.00 | 6.33 |
| Min-K% Prob[†] | 10.67 | 1.33 | 5.33 | 5.67 | 0.00 |
| Min-K%++[†] | 7.00 | 9.33 | 10.33 | 8.00 | 2.00 |
| FL | 5.67 | **10.67** | **11.00** | **8.67** | 7.00 |
| FLLoRA | 11.33 | 5.33 | 6.33 | 3.33 | 10.00 |
| IG | 5.67 | 8.00 | 10.33 | 2.33 | 11.00 |

Table 2: **White-box Setting: TPR at 3% FPR**. Comparison across all white-box methods, with the best-performing method for each metric highlighted in **bold**. † indicates baselines with grey-box access. We refer the readers to Appendix E for the complete results.

Figure 4: **White-box Setting: Our proposed attacks consistently achieve high performance**, generally outperforming the considered baselines.

addition, we include two **grey-box** attacks designed for LLMs: Min-K% Prob (Shi et al., 2023) and Min-K%++ (Zhang et al., 2024), assuming access to token-level probabilities of the generated answer to compute the membership score of each document. See Appendix B for detailed descriptions.

# 6 EVALUATION

## 6.1 WHITE-BOX SETTING

**Baseline Performance Evaluation.** Table 1 (*right*) shows the performance of baseline attacks in the white-box setting. LOSS-TA, akin to the thresholding loss attack in (Yeom et al., 2018), performs poorly on complex DocVQA models, achieving under 60% accuracy for most targets. In contrast, SCORELOSS-UA$_{all}$, which combines utility scores and loss features, achieves stronger results: 81% F1 on Donut, 75% on VT5, and 69% on Pix2Struct on DocVQA dataset. However, it underperforms LOSS-TA on PFL-DocVQA, with a 3% drop in Accuracy and 8% in F1, likely due to high loss variance in this dataset. GRADIENT-UA, which incorporates one-step gradient information, matches the performance of score-based attacks, suggesting that the gradient serves as a useful signal for membership inference. However, none of the baselines generalizes well across all target models.

**Our Proposed Attacks outperform the Baselines.** We evaluate our attacks—FL, FLLoRA, and IG—in the white-box setting against different targets. As shown in Figure 4, our methods consistently achieve high performance, indicating that *optimization-based features generalize well* across various models. Compared to all baselines, our attacks achieve either the best or near-best performance on both target datasets, with notable F1 scores of 72% against VT5 and Pix2Struct, and 82.5% against Donut. Against GRADIENT-UA, our proposed features yield up to a 10% improvement in F1 on Donut, indicating that *single-step gradients are insufficient* for reliable membership inference.

From Table 2, our attacks consistently perform well in the low-FPR regime, often surpassing or matching the strongest baselines. For instance, FL achieves a TPR of 8.67% at 3% FPR against VT5 on PFL, despite minimal overfitting, and a TPR of 11.00% against Pix2Struct-B on DocVQA. Additionally, our methods outperform both Min-K% Prob (Shi et al., 2023) and Min-K%++ (Zhang et al., 2024) across target models, underscoring their effectiveness, particularly in DocMIA setting. These results highlight the privacy risks posed by optimization-based features in membership inference.

**Why are Our Attacks more effective?** We evaluate our optimization-based features against standard metrics from prior MI attacks such as loss and single-step gradient. The loss-based attack LOSS-TA (Yeom et al., 2018) assumes that member documents exhibit lower loss values than non-members after training the target model $\mathcal{F}_t$. While this exploits the generalization gap, it is overly simplistic for large-scale models that are trained with techniques to minimize overfitting. Consequently, the strong generalization capability of DocVQA models often reduces the sensitivity of the loss as a membership indicator. In contrast, our attacks leverage the optimization process with respect to the model parameters, conditioned on each question-answer pair. We hypothesize that the *distance* resulting from parameter updates (Eq. 3) is smaller for member documents than for non-members, as depicted in Figure 2. This fine-grained signal, which captures the model's internal response to optimization, provides a more discriminative membership feature.

| Proxy Model | VT5 | | | | | |
|---|---|---|---|---|---|---|
| **Black-box** | FL | | FLLoRA | | IG | |
| | ACC | F1 | ACC | F1 | ACC | F1 |
| **PFL** VT5 | $63.33_{0.0}(+2.33)$ | $69.51_{0.03}(+0.71)$ | $63.33_{0.0}(+2.33)$ | $69.01_{0.0}(+0.21)$ | $62.00_{0.16}(+1)$ | $69.35_{0.2}(+0.55)$ |
| Donut | $70.83_{0.0}(-0.34)$ | $76.64_{0.0}(+0.87)$ | $70.83_{0.0}(-0.34)$ | $76.70_{0.0}(+0.93)$ | $70.67_{0.0}(-0.5)$ | $76.72_{0.0}(+0.95)$ |
| **DVQA** VT5 | $74.33_{0.01}(-0.84)$ | $75.08_{0.0}(-1.1)$ | $74.33_{0.0}(-0.84)$ | $74.67_{0.0}(-1.51)$ | $73.83_{0.08}(-1.34)$ | $75.81_{0.0}(-0.37)$ |
| Donut | $81.67_{0.0}(+1.17)$ | $82.54_{0.0}(+1.44)$ | $81.17_{0.0}(+0.67)$ | $82.09_{0.0}(+0.99)$ | $80.17_{0.0}(-0.33)$ | $81.89_{0.0}(+0.79)$ |
| Pix2Struct-B | $70.17_{0.0}(+1.04)$ | $69.71_{0.0}(-1.65)$ | $70.27_{0.23}(+1.14)$ | $70.85_{0.07}(-0.51)$ | $71.17_{0.0}(+2.04)$ | $72.14_{0.0}(+0.78)$ |
| Pix2Struct-L | $71.67_{0.01}(+0.84)$ | $72.13_{0.0}(+1.30)$ | $70.17_{0.0}(-0.66)$ | $71.27_{0.0}(+0.44)$ | $71.00_{0.05}(+0.17)$ | $73.15_{0.0}(+2.32)$ |

| Proxy Model | Donut | | | | | |
|---|---|---|---|---|---|---|
| **PFL** VT5 | $61.73_{0.08}(+0.73)$ | $64.04_{0.10}(-4.76)$ | $61.67_{0.08}(+0.67)$ | $63.49_{0.0}(-5.31)$ | $55.17_{0.17}(-5.83)$ | $57.37_{0.3}(-11.43)$ |
| Donut | $72.17_{0.0}(+2.33)$ | $76.24_{0.0}(-0.19)$ | $72.67_{0.0}(+1.5)$ | $77.47_{0.0}(+1.7)$ | $74.50_{0.0}(+3.33)$ | $76.43_{0.0}(+0.66)$ |
| **DVQA** VT5 | $73.50_{0.0}(-4.34)$ | $75.58_{0.0}(-4.36)$ | $74.17_{0.0}(-1)$ | $76.04_{0.0}(-0.14)$ | $74.0_{0.0}(-1.17)$ | $75.93_{0.01}(-0.25)$ |
| Donut | $79.50_{0.0}(-1)$ | $81.50_{0.0}(+0.4)$ | $80.0_{0.0}(-0.5)$ | $81.82_{0.0}(+0.72)$ | $80.27_{0.0}(-0.23)$ | $81.96_{0.0}(-0.86)$ |
| Pix2Struct-B | $70.83_{0.0}(+3.04)$ | $71.82_{0.0}(+4.88)$ | $70.83_{0.06}(+1.70)$ | $71.73_{0.14}(+0.37)$ | $71.0_{0.01}(+1.87)$ | $71.94_{0.0}(+0.58)$ |
| Pix2Struct-L | $70.83_{0.0}(0)$ | $72.95_{0.0}(+2.12)$ | $71.0_{0.0}(+0.17)$ | $72.98_{0.0}(+2.15)$ | $71.0_{0.03}(+0.17)$ | $72.81_{0.01}(+1.98)$ |

Table 3: **Black-Box Setting: Main Results of Black-Box DocMIA using Donut and VT5 as proxy models**. The checkpoints for the **black-box models** are trained on the respective datasets. Values in parentheses indicate the improvement (positive/negative) compared to the *best* number from SCORE-UA-based baselines. Results are reported over five random seeds.

| | | DVQA | | | | PFL | |
|---|---|---|---|---|---|---|---|
| | | VT5 | Donut | P2S-B | P2S-L | VT5 | Donut |
| VT5 | SCORE-TA | 9.33 | 11.00 | 8.00 | **9.00** | 5.00 | 2.67 |
| | SCORE-UA | 7.67 | 15.67 | 6.33 | 6.67 | 3.33 | 3.33 |
| | SCORE-UA_all | 9.33 | 11.00 | 8.00 | 9.00 | 5.00 | 2.67 |
| | FL | **12.33** | **23.00** | **16.67** | 5.33 | 2.00 | **8.00** |
| | FLLoRA | 11.33 | 16.33 | 9.33 | 4.67 | 3.33 | 2.00 |
| | IG | 8.33 | 7.00 | 7.67 | 7.00 | 3.67 | 6.67 |
| Donut | FL | 6.33 | 4.00 | 4.67 | 7.33 | 1.33 | 4.00 |
| | FLLoRA | 6.33 | 5.00 | 6.33 | 8.00 | 5.33 | 5.33 |
| | IG | 5.00 | 11.00 | 9.33 | 6.33 | **6.33** | 4.33 |

Table 4: **Black-box Setting: TPR at 3% FPR using Donut and VT5 as proxy models**. Comparison across all black-box methods, with the best-performing method for each metric highlighted in **bold**. The complete results can be found in the Appendix E.

As illustrated Figure 8, our *distance* feature better separates between members and non-members compared to loss-based methods (Figure 7 (*top*)). The t-SNE visualization (van der Maaten & Hinton, 2008) from Figure 7 (*bottom*) further show that features derived from our attacks create more distinct clusters in high-dimensional space across all target models, outperforming loss-based approaches. For further analysis on the impact of our selected features, see Appendix F.

## 6.2 BLACK-BOX SETTING

**Baseline Performance Evaluation.** Table 1 (*left*) presents the results of our black-box baseline attacks, all of which rely on the DocVQA score as the only source of information in this setting. Similar to the loss metric, the score metric is directly correlated with the generalization gap, making attacks more effective when there is a higher degree of overfitting. This trend is illustrated in Figure 9, where we observe strong MI performance, particularly for the Donut model with 75% in PFL and 79% F1 score in DocVQA. Meanwhile, both SCORE-UA-based baselines (Tito et al., 2024) show comparable performance, especially effective against models trained on DocVQA. Overall, no single method emerges as the clear winner across all target models.

**Attacking via Proxy Model.** Table 3 and Table 4 present the key results of our black-box attacks.

First, we observe a clear advantage of attacking the proxy models distilled with our proposed techniques. Across a wide range of black-box architectures trained on both target datasets, attacks leveraging the proxy models outperform the black-box baselines in most cases, demonstrating better MI performance. This suggests that, even without knowledge of the black-box model architecture, *one chosen proxy model still effectively distills certain behaviors* from the black-box models which are membership-indicative, enabling our attacks to infer membership with high accuracy.

When the black-box model architecture *matches* that of the proxy, we consistently observe improvements in MI performance, especially when targeting the PFL-DocVQA dataset. Among the target, *Pix2Struct is the most vulnerable* (both Base and Large). Both VT5 and Donut proxies gains of +3.04% in Accuracy and +4.88% in F1 score over the best baseline, even against the Pix2Struct-L model, which exhibits strong generalization and a minimal Train-Test gap (Figure 9). Furthermore, proxy VT5 can achieve TPRs of 23.00% and 16.67% against Donut and Pix2Struct-B, respectively, at 3% FPR on DocVQA. We also provide an analysis of the proxy model in Appendix F.3.

These results suggest that privacy vulnerabilities can be exploited using simple distillation-based strategies applied to the model's output space.

## 7 CONCLUSION

In this paper, we introduce the first document-level membership inference attacks for DocVQA models, highlighting privacy risks in multi-modal settings. By leveraging model optimization techniques, we extract discriminative features that address challenges posed by multi-modal data, repeated document occurrences in training, and auto-regressive outputs. This enables us to propose novel, auxiliary data-free attacks for both white-box and black-box scenarios. Our methods, evaluated across multiple datasets and models, outperform existing membership inference baselines, emphasizing the privacy vulnerabilities in DocVQA models and the urgent need for stronger privacy safeguards.

ACKNOWLEDGMENT

This work has been funded by the European Lighthouse on Safe and Secure AI (ELSA) from the European Union's Horizon Europe programme under grant agreement No 101070617. Views and opinions expressed are however those of the authors only and do not necessarily reflect those of the European Union or European Commission. Neither the European Union nor the European Commission can be held responsible for them. Khanh Nguyen and Dimosthenis Karatzas have been supported by the Consolidated Research Group 2021 SGR 01559 from the Research and University Department of the Catalan Government, and by project PID2023-146426NB-100 funded by MCIU/AEI/10.13039/501100011033 and FSE+.

ETHICS STATEMENT

Our research introduces two novel membership inference attacks on DocVQA models, designed to evaluate the privacy risks inherent in such systems. While our methodology exposes vulnerabilities that could potentially be exploited for malicious purposes, the primary objective of this work is to raise awareness about privacy issues in AI systems, specifically in the context of DocVQA models, and to encourage the development of more privacy-preserving technologies.

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

## APPENDIX

## A  DOCVQA DATASETS

**DocVQA** (Mathew et al., 2021) This dataset contains high-quality human annotations and is widely used as a benchmark for document understanding. It comprises of real-world administrative documents across a diverse range of types, including letters, invoices, and financial reports.

**PFL-DocVQA** (Tito et al., 2024) A large-scale dataset of real business invoices, often containing privacy-sensitive information such as payment amounts, tax numbers, and bank account details. This dataset is specifically designed for DocVQA tasks in a federated learning and differential privacy setup, supporting different levels of privacy granularity. The dataset is accompanied by a variant of MI attacks, where the goal is to infer the membership of the invoice's owner (i.e., the provider) from a set of their invoices that were not used during training.

| Split | DocVQA | | PFL-DocVQA | |
|---|---|---|---|---|
| | Num. Docs | Num. Questions | Num. Docs | Num. Questions |
| Train | 69894 | 221316 | 10194 | 39463 |
| Val | 9150 | 30491 | 1286 | 5349 |
| Test | 13463 | 43591 | 1287 | 5188 |

Table 5: **Statistics** from PFL and DocVQA dataset.

In Table 5, we present statistics for both the DocVQA and PFL-DocVQA datasets. Additionally, Figure 5 shows the distribution of questions per document: (1) while a small subset of documents have more than 10 questions, most contain fewer, and (2) a fraction of documents have only a single question. These trends hold across both datasets.

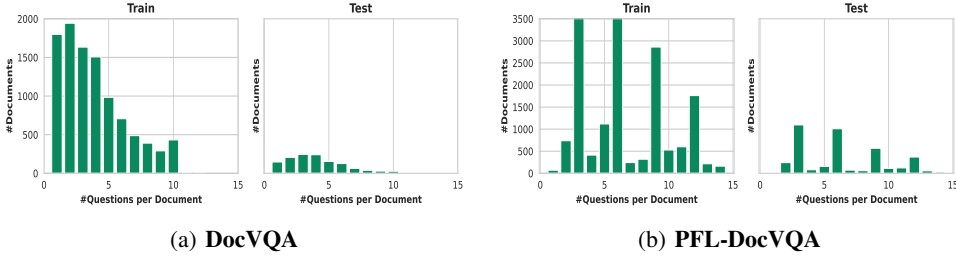

| (a) **DocVQA** | (b) **PFL-DocVQA** |

Figure 5: **The distribution of number per-document questions** from PFL and DocVQA dataset.

## B BASELINES

For the *black-box* setting, we evaluate three MI attacks as baselines, which only requires generated text to infer the membership of the target document:

**Score-Threshold Attack (SCORE-TA)** assumes that training documents should achieve higher scores than non-training ones. This attack, adapted from Yeom et al. (2018), evaluates the prediction $\hat{a}$ for each question $q$ using the utility function $\mathcal{U}$ and computes the average score $\bar{u}$. A document is then predicted as a member $\bar{u} \geq \kappa$, and non-member otherwise. The threshold $\kappa$ is set as the average value of $\bar{u}$ across $D_{\text{test}}$.

**Unsupervised Score-based Attack (SCORE-UA)** (Tito et al., 2024). This attack applies an unsupervised clustering algorithm over the set of average score $\bar{u}$ from test documents in $D_{\text{test}}$, documents within the cluster with higher average score are predicted as members.

**Unsupervised Score-based Attack - An Extension (SCORE-UA$_{\text{all}}$).** This attack extends SCORE-UA by considering multiple aggregation functions $\Phi_{\text{all}}$ to form the feature vector.

For the *grey-box* setting, we consider two additional baselines which assumes access to token-level probabilities of the generated answers $a$ to compute the membership score of each document:

**Min-K% Prob** (Shi et al., 2023) computes the average log probability of the lowest-K% answer tokens as the membership score: Min-K% Prob $= \frac{1}{|\text{Min-K\%}(a)|} \Sigma_{a_i \in \text{Min-K}(a)} \log p(a_i|a_{<i})$. Intuitively, training documents are less likely to contain low-probability answer tokens, resulting in higher scores.

**Min-K%++** (Zhang et al., 2024) also averages scores from the lowest-K% probability tokens but assumes that tokens in the predicted answers for training documents have high probabilities or often form the mode of the conditional distribution. Thus, for each token, the score is computed as: Min-K%++$(a_{<i}, a_i) = \frac{\log p(a_i|a_{<i}) - \mu_{a_{<t}}}{\sigma_{a_{<t}}}$ with $\mu_{a_{<t}}$ and $\sigma_{a_{<t}}$ are the expectation and standard deviation of $p(a_i|a_{<i})$ respectively.

We adapt these baselines to DocMIA by using an AVG aggregation function to combine scores across question-answer pairs within a document. We evaluated $K \in [0.6, 0.7, 0.8, 0.9, 1.0]$, which correspond to corresponds to 60% to 100% the length of the answer and reported the best result.

In the *white-box* setting, where loss information is available, we consider three additional baselines:

**Loss-Threshold Attack (LOSS-TA)** (Yeom et al., 2018) Similar to SCORE-TA, this attack computes the average loss $\bar{l} = \frac{1}{M} \Sigma_i^M \mathcal{L}(\mathcal{F}(x, q_i))$. A document is predicted as a member if $\bar{l} \leq \kappa$ and otherwise non-member, where $\kappa$ is selected as the average value of $\bar{l}$ across $D_{\text{test}}$.

**Unsupervised One-step Gradient Attack (GRADIENT-UA)** Inspired from Nasr et al. (2019), this attack utilizes the average norm of the gradient of the loss $\nabla_{\theta}\mathcal{L}$ from a single optimization step. It also incorporates the average score $\bar{u}$, both aggregated with $\Phi_{\text{all}}$ as the features to perform clustering.

**Unsupervised Score+Loss Attack (SCORELOSS-UA$_{\text{all}}$)** This attack extends SCOREUA$_{\text{all}}$, combining the average loss $\bar{l}$ with the average utility score $\bar{u}$, then aggregating with $\Phi_{\text{all}}$.

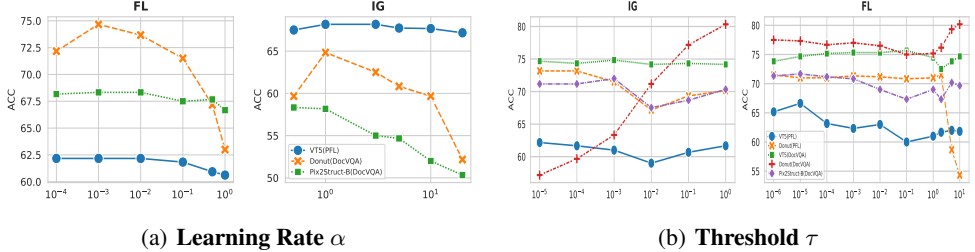

(a) **Learning Rate $\alpha$**    (b) **Threshold $\tau$**

Figure 6: **Ablation Study on Learning Rate $\alpha$ and Threshold $\tau$.** The best value for each model across all datasets is used as the hyperparameters in our black-box attacks.

| Layer | VT5(PFL) | Donut(DocVQA) | Pix2Struct-B(DocVQA) |
|---|---|---|---|
| Embedding Projection Layer | 67.0 | 71.33 | 68.66 |
| Embedding Layer Norm | 65.33 | 76.0 | 64.67 |
| Last Decoder Block FC1 | **68.33** | **78.0** | 68 |
| Last Decoder Block FC2 | 68.17 | 77.33 | **68.83** |
| Last Decoder Block Layer Norm | 61.83 | 76.83 | 67.5 |
| Random Decoder Block FC1 | 61.33 | 72.0 | 67.5 |
| Random Decoder Block FC2 | 64.0 | 73.0 | 65.17 |

Table 6: **Effect of selected layer to tune** from each target model. Attack performances are reported in terms of Accuracy.

## C    ABLATION STUDY

In this section, we provide a detailed analysis of the hyperparameter tuning process for DocMIA in the white-box setting, targeting all the considered models. Given the high computational cost due to the numerous factors involved, we focus on the key parameters that may potentially affect the attack performance. Our intuition behind this tuning process is that: achieving a reliable estimate of the distance $\Delta$ requires the optimization process to converge effectively, which in turn correlates with higher attack accuracy. Thus in all of our experiments, to increase the likelihood of convergence, we set the maximum number of optimization steps to $S = 200$. We fix the maximum number of questions per document $M$ to 10.

**Learning Rate $\alpha$.** We first study the effect of $\alpha$, which controls the speed of the optimization process in our attacks. This threshold $\tau$ is empirically set to be the average loss change observed when performing one optimization step after reaching the correct answer. Only the distance $\Delta$ and the number of steps $s$ are used as the features. For FL and FLLoRA attacks, we perform a hyperparameter search over a grid of learning rates, $\alpha \in \{10^{-4}, 0.001, 0.01, 0.1, 0.5, 1.0\}$, and $\alpha \in \{0.001, 0.01, 0.1, 0.5, 1.0, 5.0, 10.0, 20.0\}$ for the IG attacks. For FL and FLLoRA, we specifically tune the embedding projection layer, which projects the final hidden states into the vocabulary space, a common design choice across all the target models considered.

As shown in Figure 6(a), setting a high learning rate can cause the optimization process to overshoot, while lower values lead to a more stable but slower convergence. We find that a learning rate of $\alpha = 10^{-3}$ consistently delivers the best attack performance across most of the settings.

**The layer to tune L**. We now investigate the impact of layer selection on the performance of our FL and FLLoRA attacks. All target models in our study follow the transformer encoder-decoder architecture (Vaswani, 2017), where each component consists of a stack of attention layers, and a shared embedding projection layer maps the hidden states to logit vectors for prediction. Given this common structure, we examine the effect of tuning similar layers across all models, with results for attack accuracy presented in Table 6.

Our findings reveal that *layers closer to the final output exhibit higher privacy leakage* in terms of MI compared to (randomly selected) intermediate layers, likely due to receiving larger gradient updates. Specifically, fine-tuning the final fully connected layer alone leads to strong attack performance

| Model | $\alpha_{FL}$ | $\alpha_{IG}$ | $S$ | $L$ | $\tau_{FL}$ | $\tau_{IG}$ |
|---|---|---|---|---|---|---|
| VT5 | | 1.0 | | | $10^{-6}$ | $10^{-5}$ |
| Donut | 0.001 | 0.001 | 200 | last FC layer | 1.0 | 5.0 |
| Pix2Struct-B | | 0.001 | | | $10^{-4}$ | $10^{-3}$ |

Table 7: **Best Hyperaremeters from our tuning process** with consistent performance across both PFL and DocVQA dataset.

while also being more efficient in terms of the number of parameters that need to be optimized. This suggests that focusing on the last layers can achieve both high privacy leakage and computational efficiency in our MI attacks.

**Threshold $\tau$.** With the optimizer OPT and learning rate $\alpha$ fixed, the threshold $\tau$ emerges as the most critical hyperparameter that requires careful tuning for each attack. We experiment with a wide range of $\tau$ values, spanning from $10^{-6}$ to 10.0, and select the optimal value based on attack performance, as demonstrated in Figure 6(b). This optimal $\tau$ is then applied consistently in all subsequent experiments. Careful selection of this threshold is crucial, as it directly influences the stability and success of the optimization process.

We summarize the set of tuned hyperparameters for our approach in Table 7.

## D  MORE ON ATTACK IMPLEMENTATION

### D.1  TARGET MODEL TRAINING

For all target models, whenever feasible, we utilize the public checkpoint fine-tuned on the considered private dataset from Hugging Face library and adhere to the data processing guidelines, such as document resolution, as recommended by the authors. We deliberately opt for public checkpoints for two reasons: (1) to make it consistent to further research in privacy attacks that use the same trained models, and (2) to minimizing the biases in model training that affect the final results, given the complexity of the original training process and our limited resources. Table 8 summarizes the details of the process from which public checkpoints for the target models considered in this work are obtained. This includes the datasets the models were pre-trained on, before by fine-tuning on target DocVQA datasets, along with the corresponding download URLs for these checkpoints.

| Model | Num. Params | Downstream Task | Data | | Checkpoint | |
|---|---|---|---|---|---|---|
| | | | Pretrain | Finetune | Pretrain | Finetune |
| VT5 | 250M | DocVQA | C4+IIT-CDIP | PFL DocVQA | | https://benchmarks.elsa-ai.eu/?ch=2 |
| Donut | 200M | DocVQA | CDIP 11M + 0.5M synthesized Docs | PFL DocVQA | naver-clova-ix/donut-base[†] | Ours naver-clova-ix/donut-base-finetuned-docvqa[†] |
| Pix2struct-B | 282M | DocVQA | BooksCorpus + C4 Web HTML | DocVQA | google/pix2struct-base[†] | google/pix2struct-docvqa-base[†] |
| Pix2struct-L | 1.33B | | | | google/pix2struct-large[†] | google/pix2struct-docvqa-large[†] |

Table 8: **Details of the public checkpoints** used as target models in this work. † denotes checkpoint from Hugging Face.

| Model | Optimizer | Learning Rate | Weight Decay | Batch Size | Scheduler | Iteration |
|---|---|---|---|---|---|---|
| VT5 | AdamW | 2e-4 | - | 16 | - | 200k |
| Donut | Adam | 3e-5 | 0.01 | 4 | linear warmup 10% | 800k |
| Pix2Struct-B | AdaFactor | 1e-5 | - | 4 | warmup 1000 steps, cosine decay to 0 | 800k |

Table 9: **Details of the training hyperparameter** for each target model in this work.

If public checkpoints are unavailable, we fine-tune the selected model on the respective private dataset, using the pre-trained checkpoint as the initialization, with the training procedure outlined by the respective authors. To prevent overfitting, we perform early stopping based on validation performance, ensuring that all evaluated models generalize well to unseen data. We also use the pre-trained checkpoint to initialize the proxy model $\mathcal{F}_p$ to train it on $D_{query}$. We provide an overview

| Dataset | Model | Test Set | ACC | ANLS | Train-Test Gap |
|---|---|---|---|---|---|
| PFL | VT5 | Original | 81.4 | 90.17 | - |
| | | MIA | 82.74 | 90.91 | 11.44 |
| | | MIA-rephrased | 77.59 | 85.84 | - |
| | Donut | Original | 74.73 | 88.66 | - |
| | | MIA | 80.15 | 91.64 | 22.2 |
| | | MIA-rephrased | 70.46 | 80.96 | - |
| DVQA | VT5 | Original | 60.1 | 69.33 | - |
| | | MIA | 75.54 | 81.69 | 36.22 |
| | | MIA-rephrased | 73.57 | 79.89 | - |
| | Donut | Original | 59.26 | 66.91 | - |
| | | MIA | 78.55 | 83.42 | 39.78 |
| | | MIA-rephrased | 72.57 | 77.12 | - |
| | Pix2Struct-B | Original | 57.11 | 68.13 | - |
| | | MIA | 64.42 | 79.95 | 25.8 |
| | | MIA-rephrased | 63.81 | 74.06 | - |
| | Pix2Struct-L | Original | 64.53 | 74.12 | - |
| | | MIA | 73.91 | 82.71 | 22.11 |
| | | MIA-rephrased | 69.93 | 79.15 | - |

Table 10: **DocVQA Performance of the target models on PFL and DocVQA dataset.** Train-Test Gap is computed as the different of DocVQA Accuracy between *member/non-member* documents. MIA denotes the attack evaluation set, which is a subset randomly sampled from the original train/test set, MIA-rephrased is its variants with rephrased questions by LLM.

of the training procedure for each target model, based on the respective papers. These procedures were adapted to fit our computational resources, as outlined in Table 9.

## D.2 TARGET MODEL PERFORMANCE ON DOCVQA

To ensure the utility of the target models for our experiments, we validated that the DocVQA performance of each model checkpoint closely matched the results reported in the respective papers. Table 10 presents the target models' performance across both DocVQA datasets. We observe a clear train-test performance gap, particularly in smaller models, while the gap tends to narrow for more generalized models or with increased dataset size.

| | Model | SCORE-TA | | SCORE-UA | | SCORE-UA$_{all}$ | | LOSS-TA | | GRADIENT-UA | | SCORELOSS-UA$_{all}$ | | Min-K%[†] | | Min-K%++[†] | |
|---|---|---|---|---|---|---|---|---|---|---|---|---|---|---|---|---|---|
| | | ACC | F1 | ACC | F1 | ACC | F1 | ACC | F1 | ACC | F1 | ACC | F1 | ACC | F1 | ACC | F1 |
| PFL | VT5 | **62.33** | **64.13** | $61.00_{0.0}$ | $68.80_{0.0}$ | $60.67_{0.0}$ | $60.67_{0.0}$ | 57.83 | 62.81 | $60.67_{0.0}$ | $60.67_{0.0}$ | $60.67_{0.0}$ | $60.67_{0.0}$ | 57.17 | 64.84 | **61.33** | **72.16** |
| | Donut | **73.33** | **75.14** | $68.33_{0.0}$ | $75.77_{0.0}$ | $71.17_{0.67}$ | $71.33_{3.32}$ | 73.67 | 78.99 | $70.67_{0.0}$ | $69.55_{0.0}$ | $70.83_{0.0}$ | $69.67_{0.0}$ | 36.5 | 32.09 | 50.5 | 60.35 |
| DVQA | VT5 | **75.67** | **75.75** | $72.17_{0.0}$ | $76.18_{0.0}$ | $75.17_{0.0}$ | $75.13_{0.0}$ | 73.67 | 77.99 | $71.17_{0.0}$ | $67.54_{0.0}$ | $\mathbf{75.50_{0.0}}$ | $\mathbf{76.02_{0.0}}$ | 71.0 | 76.16 | 66.67 | 74.56 |
| | Donut | 79.67 | 79.53 | $75.97_{0.07}$ | $79.57_{0.07}$ | $\mathbf{80.50_{0.0}}$ | $\mathbf{81.10_{0.0}}$ | 51.83 | 53.46 | $77.17_{0.0}$ | $75.92_{0.0}$ | $\mathbf{80.50_{0.0}}$ | $\mathbf{81.10_{0.0}}$ | 47.0 | 48.38 | 53.33 | 59.89 |
| | Pix2Struct-B | 67.33 | 67.97 | $\mathbf{68.17_{0.0}}$ | $\mathbf{71.36_{0.0}}$ | $69.13_{0.07}$ | $67.67_{0.09}$ | 59.33 | 64.63 | $66.0_{0.0}$ | $68.32_{0.0}$ | $69.00_{0.0}$ | $67.48_{0.0}$ | **73.67** | **76.97** | 54.50 | 66.99 |

Table 11: **Results from Baseline Attacks..** Gray color indicate attacks conducted in the black-box setting. [†] indicates methods requiring grey-box access. Results are reported based on five random seeds for KMEANS, if any. The methods with the best *average* performance across the two metrics are highlighted in **bold**.

| | White-box | FL | | FLLoRA | | IG | |
|---|---|---|---|---|---|---|---|
| | | ACC | F1 | ACC | F1 | ACC | F1 |
| PFL | VT5 | $\mathbf{66.63_{0.07}}$**(+5.96)** | $\mathbf{72.40_{0.1}}$**(+11.73)** | $65.17_{0.0}(+4.50)$ | $70.52_{0.0}(+9.85)$ | $61.83_{0.0}(+1.16)$ | $69.18_{0.0}(+8.51)$ |
| | Donut | $72.67_{0.0}(+1.5)$ | $77.96_{0.0}(+6.63)$ | $72.83_{0.0}(+1.66)$ | $77.94_{0.0}(+6.61)$ | $\mathbf{75.17_{0.0}}$**(+4)** | $\mathbf{79.39_{0.0}}$**(+8.06)** |
| DVQA | VT5 | $75.67_{0.0}(+0.5)$ | $76.60_{0.0}(+1.47)$ | $75.57_{0.08}(+0.4)$ | $77.31_{0.13}(+2.18)$ | $74.83_{0.0}(-0.34)$ | $76.88_{0.0}(+1.75)$ |
| | Donut | $80.33_{0.0}(-0.17)$ | $82.18_{0.0}(+1.08)$ | $80.0_{0.0}(-0.5)$ | $81.93_{0.0}(+0.83)$ | $80.33_{0.0}(-0.17)$ | $82.34_{0.0}(+1.24)$ |
| | Pix2Struct-B | $71.67_{0.0}(+2.54)$ | $72.22_{0.0}(+4.55)$ | $70.50_{0.0}(+1.37)$ | $71.95_{0.0}(+4.28)$ | $\mathbf{72.00_{0.0}}$**(+2.87)** | $\mathbf{72.82_{0.0}}$**(+5.15)** |

Table 12: **White-Box: Main Results of DocMIA.** Values in parentheses indicate the improvement (positive/negative) of our proposed attacks compared to the SCORELOSS-UA$_{all}$. Compared to all baselines, the methods with the best *average* performance across the two metrics are highlighted in **bold**. Results are reported over five random seeds.

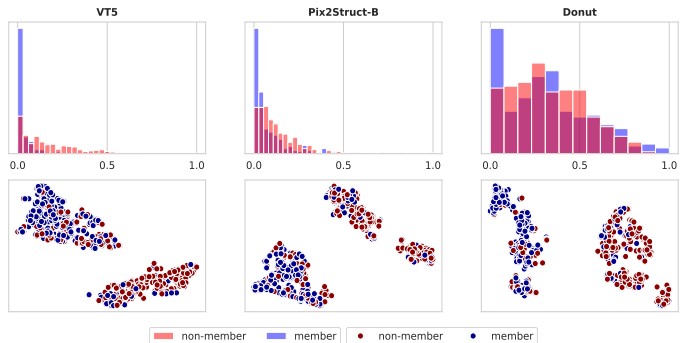

Figure 7: **Membership Features against three different target models on DocVQA Dataset.** *Top*: The distribution of average *loss* over all questions from all target documents on each target model. *Bottom*: T-SNE visualization of the features used in our proposed attacks.

Figure 8: **Distribution of Average *Distance*** from member and non-member documents.

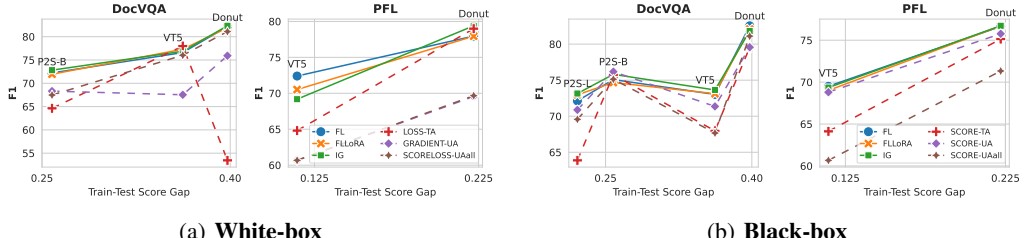

(a) **White-box**

(b) **Black-box**

Figure 9: **MI performance versus the Train-Test gap.** The target models exhibit varying Train-Test gaps, measured by the difference in DocVQA scores between member/non-member documents. Our attacks remain effective even when the gap is small, with performance improving as the gap increases across most target models and datasets. In contrast, baseline methods show more variable performance under these conditions.

### D.3 COMPUTATION AND RUNTIME

All attack methods are implemented using PyTorch and executed on an NVIDIA GeForce A40 GPU with 45 GB of memory. The maximum runtime for each attack does not exceed *10 hours* per run, depending on the target model's size and the pre-processing steps required for the data. This runtime reflects the efficiency of our approach, especially when compared to methods based on shadow training, which require retraining of large-scale models many times to be effective (Carlini et al., 2022). Our results demonstrate that the proposed attacks are both efficient and scalable, making them practical for large-scale models in real-world applications.

## E MORE ON ATTACK RESULTS

In this section, we provide the full results of all evaluated attacks in terms of Accuracy and F1 Score, summarized in Tables 11 and 12. In addition, we evaluate our attacks using TPR@1%FPR and TPR@3%FPR, following standard practices in recent MIA literature, as shown in Table 13 and 20.

An interesting observation is the high performance of the LOSS-TA method for VT5 on DocVQA and Donut on PFL in Table 13. This performance can be attributed to the clear separation in the loss distribution between member and non-member samples (Figure 7), which indicates overfitting behavior in these cases.

## F MORE ON ANALYSIS

In this section, we provide a deeper analysis of the effectiveness of our proposed white-box and black-box attacks, highlighting their performance relative to the baseline approaches.

| | DVQA | | | | | | PFL | | | |
| | VT5 | | Donut | | Pix2Struct-B | | VT5 | | Donut | |
| | 1% | 3% | 1% | 3% | 1% | 3% | 1% | 3% | 1% | 3% |
|---|---|---|---|---|---|---|---|---|---|---|
| Loss-TA | **7.67** | **14.00** | 0.67 | 7.67 | 2.33 | 5.33 | 0.67 | 3.00 | 1.67 | **14.67** |
| Gradient-UA | 2.33 | 9.33 | 3.67 | 6.00 | 1.00 | 5.00 | 0.33 | 3.00 | 1.00 | 8.33 |
| ScoreLoss-UA$_{all}$ | 1.33 | 4.67 | 2.67 | 8.67 | 2.00 | 6.67 | 0.33 | 4.00 | 0.67 | 6.33 |
| Min-K% | 2.67 | 10.67 | 0.33 | 1.33 | 0.33 | 5.33 | 1.67 | 5.67 | 0.00 | 0.00 |
| Min-K%++ | 1.00 | 7.00 | **4.33** | 9.33 | 0.67 | 10.33 | 1.00 | 8.00 | 0.33 | 2.00 |
| FL | 2.33 | 5.67 | 3.33 | **10.67** | **6.00** | **11.00** | **3.67** | **8.67** | 0.33 | 7.00 |
| FLLoRA | 3.33 | 11.33 | 2.67 | 5.33 | 3.67 | 6.33 | 1.33 | 3.33 | 0.33 | 10.00 |
| IG | 0.67 | 5.67 | 1.33 | 8.00 | 3.00 | 10.33 | 1.00 | 2.33 | **5.67** | 11.00 |

Table 13: **White-box: TPR at fixed FPR**. Comparison across all white-box methods, with the best performance highlighted in **bold**. 1% and 3% indicate TPR@1%FPR and TPR@3%FPR respectively.

## F.1 Impact of Selected Features

As outlined in the main paper, we fix the set of selected features across all experiments. These features include the DocVQA score $u$, the optimization-based distance $\Delta$, and the number of optimization steps $s$, aggregated using the set of aggregation functions $\Phi_{all} = \{\text{AVG}; \text{MIN}; \text{MAX}; \text{MED}\}$. We first evaluate the impact of individual features and their combinations on attack performance in the white-box DocMIA setting, using AVG as the aggregation function $\Phi$. The analysis employs the best hyperparameters identified during the tuning process described in Section C.

| VT5 | | | | Donut | | | |
| AVG(NLS) | AVG($\Delta$) | AVG($s$) | F1 | AVG(NLS) | AVG($\Delta$) | AVG($s$) | F1 |
|---|---|---|---|---|---|---|---|
| ✓ | | | 68.88 | ✓ | | | 67.58 |
| | ✓ | | 71.45 | | ✓ | | 71.36 |
| | | ✓ | 70.92 | | | ✓ | 73.16 |
| ✓ | ✓ | | 71.09 | ✓ | ✓ | | 72.87 |
| ✓ | | ✓ | 71.11 | ✓ | | ✓ | 73.67 |
| | ✓ | ✓ | 71.22 | | ✓ | ✓ | 73.86 |
| ✓ | ✓ | ✓ | **71.53** | ✓ | ✓ | ✓ | **73.89** |

Table 14: **Impact of Selected Features on PFL-DocVQA Models.**

| | PFL | | DVQA | | |
| | VT5 | Donut | VT5 | Donut | Pix2Struct-B |
|---|---|---|---|---|---|
| AVG($\ell$) | 67.53 | 67.80 | 73.43 | 56.79 | 69.97 |
| AVG($\|\nabla_\theta\mathcal{L}\|_2$) | 70.53 | 71.51 | 71.91 | 71.53 | 66.14 |
| AVG($\Delta$) | 71.45 | 71.36 | 72.86 | 57.34 | 70.57 |
| AVG($s$) | 70.92 | 73.16 | 74.34 | 60.32 | 69.00 |

| | PFL | | DVQA | | |
| | VT5 | Donut | VT5 | Donut | Pix2Struct-B |
|---|---|---|---|---|---|
| $\Phi = $ AVG | 71.53 | 73.89 | 74.96 | 72.94 | 73.22 |
| $\Phi = \Phi_{all}$ | 72.4(+0.87) | 77.96(+4.07) | 76.6(+1.67) | 82.18(+9.24) | 72.22(-1.0) |

Table 15: **Comparisons in Attack Performance in terms of F1 Score**: *(Top)* between our Optimization-based Features with the loss value $\ell$ (Eq. 4) and the gradient norm $\|\nabla_\theta\mathcal{L}\|_2$. *(Bottom)* between AVG and $\Phi_{all}$ as the aggregation functions.

| VT5 | | | | Donut | | | | Pix2Struct-B | | | |
| AVG(NLS) | AVG($\Delta$) | AVG($s$) | F1 | AVG(NLS) | AVG($\Delta$) | AVG($s$) | F1 | AVG(NLS) | AVG($\Delta$) | AVG($s$) | F1 |
|---|---|---|---|---|---|---|---|---|---|---|---|
| ✓ | | | 72.73 | ✓ | | | **76.88** | ✓ | | | 72.60 |
| | ✓ | | 72.86 | | ✓ | | 57.34 | | ✓ | | 70.57 |
| | | ✓ | 74.34 | | | ✓ | 60.32 | | | ✓ | 69.00 |
| ✓ | ✓ | | **75.81** | ✓ | ✓ | | 65.94 | ✓ | ✓ | | 73.20 |
| ✓ | | ✓ | 75.04 | ✓ | | ✓ | 72.17 | ✓ | | ✓ | 72.87 |
| | ✓ | ✓ | 74.19 | | ✓ | ✓ | 60.29 | | ✓ | ✓ | 70.17 |
| ✓ | ✓ | ✓ | 74.96 | ✓ | ✓ | ✓ | 72.94 | ✓ | ✓ | ✓ | **73.22** |

Table 16: **Impact of Selected Features on DocVQA Target Models**. Only AVG is used as the aggregation function $\Phi$. Attack performances are obtained with our FL method using the best hyperparameters.

Table 14 and Table 16 summarize the attack performance when individual features or their combinations are used. Additionally, Table 15 *(Top)* compares the attack performance of our optimization-based features with the loss value $\ell$ and the gradient norm of the loss with respect to the model parameters $\theta$. Here, the loss value $\ell$ is computed uniformly across all target models over $K$ generation steps, given a (document, question, answer) example $(x, q, a)$ as:

$$\ell = -\sum_{k=1}^{K} \log p_\theta(a_k | a_{<k}, x, q) \tag{4}$$

When used individually, our proposed optimization-based features outperform the DocVQA score and the loss in most cases. Our attack methods are particularly effective against target models like VT5 and Donut trained on PFL-DocVQA, which exhibit lower overfitting and small Train-Test gaps (as shown in Table10). These results highlight that our attacks provide more discriminative features than the commonly used MIA features.

When combined, our selected features achieve the best or near-best performance across all cases. Furthermore, extending aggregation functions from AVG to $\Phi_{all}$ adds notable improvements in attack effectiveness, as shown in Table 15 *(Bottom)*. These results demonstrate that our proposed feature set is robust across different target models, making it a reliable choice for DocMIA.

## F.2 IMPACT OF THE TRAINING QUESTIONS KNOWLEDGE

So far, our document MI attacks against DocVQA models have assumed complete knowledge of the original training questions. We now relax this assumption and investigate how the lack of access to the exact training questions affects attack performance. In practice, an adversary may not have access to the exact training questions but can approximate them. For example, documents like invoices often follow standard layouts, and biases in human annotation may lead to predictable patterns in the types of questions asked during the creation of DocVQA datasets (Tito et al., 2024; Mathew et al., 2021). It is important to note that the original training questions tend to be simple, natural questions designed to extract specific information from the document. Moreover, the type of question is inherently linked to the type of document on which the DocVQA model is trained. For instance, if the target model is trained on invoices, the natural type of question would focus on extracting essential details from the invoice, such as the "total amount", framed in a clear and straightforward manner e.g., "What is the total?". This makes it possible for an adversary to generate approximate versions of the training questions, simulating a more realistic attack setting.

| | Model | SCORE-TA | | SCORE-UA$_{all}$ | | LOSS-TA | | SCORELOSS-UA$_{all}$ | | OURS (FL) | |
|---|---|---|---|---|---|---|---|---|---|---|---|
| | | ACC | F1 | ACC | F1 | ACC | F1 | ACC | F1 | ACC | F1 |
| PFL | VT5 | 60.67 | 64.13 | 55.83$_{0.0}$ | 46.89$_{0.0}$ | 54.50 | 59.19 | 55.83$_{0.0}$ | 46.89$_{0.0}$ | **64.00$_{0.0}$** | **69.14$_{0.0}$** |
| | Donut | 69.17 | 69.72 | 59.33$_{0.0}$ | 51.59$_{0.0}$ | 68.50 | 66.67 | 59.17$_{0.0}$ | 51.49$_{0.0}$ | **71.13$_{0.08}$** | **72.07$_{0.0}$** |
| DVQA | VT5 | 73.67 | 75.01 | 74.83$_{0.0}$ | 74.36$_{0.0}$ | 71.67 | 74.06 | 75.17$_{0.0}$ | 74.96$_{0.0}$ | **74.83$_{0.0}$** | **75.68$_{0.0}$** |
| | Donut | **69.17** | **71.23** | 65.17$_{0.0}$ | 62.21$_{0.0}$ | 52.33 | 53.57 | 65.17$_{0.0}$ | 62.21$_{0.0}$ | 67.67$_{0.0}$ | 68.51$_{0.0}$ |

Table 17: **Results with Rephrased Questions.** Gray color indicate attacks conducted in the black-box setting. All results are reported based on five random seeds. The methods with the best *average* performance across the two metrics are highlighted in **bold**.

| | DVQA | | | | PFL | | | |
|---|---|---|---|---|---|---|---|---|
| | VT5 | | Donut | | VT5 | | Donut | |
| | 1% | 3% | 1% | 3% | 1% | 3% | 1% | 3% |
| Min-K% | 3.00 | 4.33 | 0.33 | 1.00 | **6.33** | **20.33** | 2.00 | 2.33 |
| Min-K%++ | 3.00 | 4.67 | 0.00 | 2.67 | 6.33 | 10.00 | 0.00 | 7.00 |
| FL | 0.67 | 5.00 | **3.33** | **8.00** | 3.67 | 17.33 | 3.00 | 4.67 |
| FLLoRA | **5.00** | **9.33** | 0.67 | 3.67 | 5.00 | 9.33 | **4.33** | **10.00** |
| IG | 5.33 | 8.00 | 1.00 | 5.00 | 5.33 | 8.00 | 1.67 | 10.00 |

| | PFL | | DVQA | | | |
|---|---|---|---|---|---|---|
| | VT5 | | Donut | | Pix2Struct-B | |
| | 1% | 3% | 1% | 3% | 1% | 3% |
| SCORE-TA | 0.33 | 2.67 | **3.33** | 9.67 | 3.00 | 8.67 |
| SCORE-UA$_{all}$ | 0.33 | 2.67 | 2.33 | 9.33 | **4.67** | 8.67 |
| FL | 0.33 | 1.33 | 0.33 | 4.00 | 1.33 | 4.67 |
| FLLoRA | 1.00 | 5.33 | 1.67 | 5.00 | 2.33 | 6.33 |
| IG | **2.67** | **6.33** | 1.67 | **11.00** | 3.67 | **9.33** |

Table 18: **White-box Results: TPR at 1% and 3% FPR with Rephrased Questions**. Comparison to *white-box* methods: Min-K% and Min-K%++ methods, with the best method in **bold**.

Table 19: **Black-box Results: TPR at 1% and 3% FPR with Rephrased Questions**. Donut is used as The Proxy Model.

To explore this scenario, we conduct experiments where we paraphrase the original training questions using Mistral (Jiang et al., 2023), and use these rephrased questions as inputs for the MI attacks. As illustrated in Table 17, the performance of all MI attacks declines when rephrased questions are used, mirroring the drop in DocVQA model performance (Table 10), which is expected due to the increased uncertainty introduced by question rephrasing.

Among the baselines, the SCORE-TA attack proves particularly to be robust, especially against models trained on DocVQA, which show a higher degree of overfitting. In contrast, attacks incorpo-

| Target | | DVQA | | | | | | | | PFL | | | |
|---|---|---|---|---|---|---|---|---|---|---|---|---|---|
| | | VT5 | | Donut | | Pix2Struct-B | | Pix2Struct-L | | VT5 | | Donut | |
| Proxy | | 1% | 3% | 1% | 3% | 1% | 3% | 1% | 3% | 1% | 3% | 1% | 3% |
| | SCORE-TA | 4.00 | 9.33 | 5.00 | 11.00 | **5.33** | 8.00 | 3.33 | **9.00** | 1.00 | 5.00 | 0.67 | 2.67 |
| | SCORE-UA | 3.67 | 7.67 | 4.33 | 15.67 | 4.00 | 6.33 | 4.33 | 6.67 | 0.67 | 3.33 | 0.33 | 3.33 |
| | SCORE-UA$_{all}$ | 4.00 | 9.33 | 5.00 | 11.00 | 5.33 | 8.00 | 3.33 | 9.00 | 1.00 | 5.00 | 0.67 | 2.67 |
| | FL | 0.67 | **12.33** | **11.67** | **23.00** | 2.00 | **16.67** | 2.00 | 5.33 | 0.67 | 2.00 | **5.00** | 8.00 |
| VT5 | FLLoRA | **4.67** | 11.33 | 6.34 | 16.33 | 2.33 | 9.33 | 1.00 | 4.67 | 2.00 | 3.33 | 0.00 | 2.00 |
| | IG | 1.00 | 8.33 | 2.00 | 7.00 | 4.67 | 7.67 | 2.33 | 7.00 | 0.33 | 3.67 | 1.33 | 6.67 |
| | FL | 0.33 | 6.33 | 0.33 | 4.00 | 1.33 | 4.67 | 3.00 | 7.33 | 0.33 | 1.33 | 1.33 | 4.00 |
| Donut | FLLoRA | 1.00 | 6.33 | 1.67 | 5.00 | 2.33 | 6.33 | 3.00 | 8.00 | 0.00 | 5.33 | 2.00 | 5.33 |
| | IG | 1.67 | 5.00 | 0.67 | 11.00 | 3.67 | 9.33 | **4.67** | 6.33 | **2.67** | **6.33** | 1.67 | 4.33 |

Table 20: **Black-box: TPR at fixed FPR**. Comparison across all black-box methods, with the best-performing method highlighted in **bold**. 1% and 3% indicate TPR@1%FPR and TPR@3%FPR respectively.

rating loss-based signals introduce additional noise due to uncertainty, leading to a noticeable drop in performance.

Despite the rephrasing, our attacks remain effective, maintaining performance levels comparable to those observed with the original questions, especially against the two PFL models, which demonstrate a lower degree of overfitting.

We also evaluate our proposed attacks against other methods in this setting, focusing on TPR at 1% and 3% FPR, with the results summarized in Table 18 and 19.

### F.3 THE RESULTING PROXY MODEL

The purpose of training the Proxy Model on $D_{query}$, with labels generated by the black-box model, is to mimic the prediction patterns of the black-box model. The expectation is that the proxy model can capture internal decision-making patterns by following the black-box's prediction strategies. Instead of optimizing for ground-truth labels, we train the proxy to maximize the likelihood of the generated labels. The training process concludes when the proxy achieves near-zero training loss, at which point the final checkpoint is used for the attack.

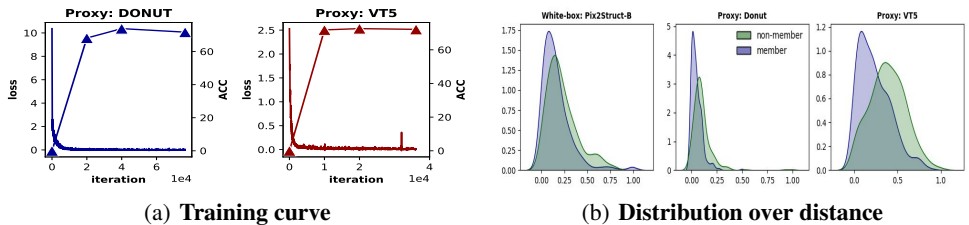

(a) **Training curve**          (b) **Distribution over distance**

Figure 10: **The resulting Proxy Model** against Pix2Struct-B in the black-box setting. (*a*) The attack accuracy improves quickly once the loss reaches near zero. (*b*) The optimization distance values between member and non-member documents exhibit a separation similar to that seen in the white-box setting.

As illustrated in Figure 10(a), the attack performance quickly improves as training progresses. The model overfits quickly, with attack performance reaching its peak early—after just a quarter of the training process—demonstrating the efficiency of our approach. This suggests that *once the proxy model converges, it has effectively captured informative membership signals from the black-box model*, making it ready for the attack. Moreover, we compare the distribution of optimization distances between the proxy model and the same model in the white-box setting, as shown in Figure 10(b). The results show a similar degree of separation between the two clusters in both cases, indicating the proxy model's effectiveness in approximating the black-box model's behavior to a certain extent.

## F.4 Attack Performance against Minimal-Training Documents

DocVQA models typically process each question-answer pair independently, resulting in multiple exposures of each document during training. This increases the likelihood of being memorized by the model, making such documents more vulnerable to MIAs. Intuitively, documents associated with fewer training questions should be less exposed and therefore be less vulnerable.

| | Model | FL | | | | IG | | | |
|---|---|---|---|---|---|---|---|---|---|
| | | $m = 1(1)$ | $m = 2(1)$ | $m = 3(85)$ | ALL(300) | $m = 1(1)$ | $m = 2(1)$ | $m = 3(85)$ | ALL(300) |
| **PFL** | VT5 | 0 | 0 | 83.53 | 87.67 | 100 | 100 | 85.88 | 86.33 |
| | Donut | 100 | 100 | 100 | 97.67 | 100 | 100 | 97.65 | 97 |
| | Model | $m = 1(51)$ | $m = 2(60)$ | $m = 3(52)$ | ALL(300) | $m = 1(51)$ | $m = 2(60)$ | $m = 3(52)$ | ALL(300) |
| **DVQA** | VT5 | 86.27 | 71.67 | 84.62 | 77.00 | 90.2 | 85 | 86.54 | 80.67 |
| | Donut | 88.24 | 73.33 | 76.92 | 77.33 | 56.86 | 68.33 | 55.77 | 61.33 |
| | Pix2Struct-B | 90.2 | 93.33 | 90.38 | 87 | 88.24 | 88.33 | 76.92 | 73 |

Table 21: **Membership Prediction Accuracy on *Member* Documents with minimal repetition.** $m$ denotes the subset of testing documents with $m$ *training* questions, with subset sizes shown in parentheses. Compared to the performance measured on the entire member set (denoted as ALL), our attacks are still robust against documents with the low risk of memorization.

To evaluate this, we measure the accuracy of membership predictions from our attacks on a subset of *member* documents in $D_{\text{test}}$ associated with only a few training questions. These documents represent a minimal memorization risk, posing a more challenging evaluation scenario. Results in Table 21 show that our attacks remain effective on these subsets, achieving high accuracy even for documents $m = 1$ training question. This demonstrates the robustness of our attacks under conditions of minimal repetition.

## G Defenses

To mitigate the privacy vulnerabilities associated with membership inference attacks in Document Visual Question Answering (DocVQA) systems, we can employ Differential Privacy (DP) techniques (Dwork et al., 2014), specifically through the use of differentially private stochastic gradient descent (DP-SGD) introduce by Abadi et al. (2016). DP is a robust framework that ensures an individual's data contribution cannot be inferred, even when an adversary has access to the model's outputs. DP-SGD achieves this by adding calibrated noise to the model's gradients during training, thus providing strong theoretical privacy guarantees. However, this approach is not without its drawbacks; the necessity of noise injection can adversely affect the utility of the trained model, leading to reduced performance in answering queries accurately. Alternatively, we can consider ad-hoc solutions such as limiting the number of queries to one question per document in black-box setting, which would inherently reduce the model's usability and flexibility in practical applications. While these measures can enhance privacy, they also necessitate careful consideration of the balance between privacy protection and the functionality of DocVQA systems.

| | $\varepsilon = 8$ | | | $\varepsilon = 32$ | | | $\varepsilon = \infty$ | | |
|---|---|---|---|---|---|---|---|---|---|
| | ANLS | F1 | TPR@3%FPR | ANLS | F1 | TPR@3%FPR | ANLS | F1 | TPR@3%FPR |
| FL | | 55.09 | 2.33 | | 58.84 | 4.33 | | 73.81 | 7.33 |
| FLLoRA | 19.16 | 54.94 | 2.00 | 21.81 | 58.94 | 3.67 | 50.12 | 73.81 | 7.33 |
| IG | | 56.29 | 1.67 | | 59.35 | 5.00 | | 73.52 | 8.67 |

Table 22: **DocMIA Results for Donut trained with DP-SGD on DocVQA dataset**. We report the attack performance of our FL method in terms of F1 score and TPR3%FPR.

To evaluate the robustness of our proposed membership inference attacks against Differential Privacy (DP), we implemented the well-known DP-SGD algorithm. We considered five privacy budget $\varepsilon \in \{8, 32\}$, with corresponding noise multiplier $\sigma \in \{0.5767822266, 0.3824234009\}$, respectively. The composition of the privacy budget over multiple iterations was calculated using Rényi Differential Privacy (RDP). We then converted the RDP guarantees into the standard $(\varepsilon, \delta)$-DP notion following the conversion theorem from (Balle et al., 2020).

We trained the Donut model on the DocVQA dataset with DP-SGD to provide theoretical privacy guarantees for individual training documents. Due to resource constraints, we resized document resolution to a smaller size (1280, 960) compared to (2560, 1920) in the public checkpoint provided by the original authors, which slightly reduced the model's DocVQA performance. For additional details on the effects of document resolution, we refer readers to the original model's paper(Kim et al., 2022). The model was trained using the Adam optimizer with a learning rate of $1e-4$, for 10 epochs, and with a batch size of 16. DocVQA performance was evaluated using the Average Normalized Levenshtein Similarity (ANLS) metric.

Table 22 summarizes the results. As expected, introducing DP into model training significantly reduces the attack performance, for example from 73.81% F1 score with non-DP model to 55.09% at $\varepsilon = 8$, but this comes at the cost of substantial utility degradation, with the DP model achieving less than half of the performance of the non-DP model, 21.81 of ANLS at $\varepsilon = 8$ compared to 50.12 of ANLS from non-DP checkpoint. For higher privacy budgets ($\varepsilon = 32$), our attacks demonstrate improved effectiveness, achieving notable gains, +3.75 in F1 and +2 in TPR3%FPR scores compared to $\varepsilon = 8$, as the model becomes less privacy-constrained.

