# OpenReview forum: "DocMIA: Document-Level Membership Inference Attacks against DocVQA Models"
_ICLR.cc/2025/Conference — ICLR 2025 Poster_

### Official Review · Reviewer_V2Bv · 2024-10-29

**Soundness:** 3
**Presentation:** 3
**Contribution:** 3
**Rating:** 6
**Confidence:** 5

**Summary:**

This work tackles document-level inference for VQA models that have been trained on text as well as visual-documents, where additional processing steps like OCR and repeated instances of documents make the problem significantly different from standard MI under language models. The authors propose attacks that are free of any auxiliary data, for both black-box and white-box access models.  The proposed attacks are evaluated on multiple datasets, outperforming both existing and newly-designed baseline inference methods.

**Strengths:**

- The work tackles document VQA systems, which post a lot more challenges than simple text-based language models. Understanding leakage in this scenario would be helpful.
- The attack methodology is interesting and straightforward, and the authors experiment with multiple ways of "fine-tuning" the target model to measure scores used for membership inference.
- Results are presented very well, and overall the figures/tables are very neat, with a very detailed Appendix.

**Weaknesses:**

I think the works make a fair contribution, but I do have some concerns (major and minor listed below and under questions) that the authors could answer to help me better understand some of their design choices.

- The evaluation design is completely disconnected from how MI attacks are usually evaluated. Currently, thresholds are computed based on average values in $D_\text{test}$. In MI (or similar evaluations), the standard protocol is to sweep over the non-member's values and based on the FPR (false positive rate) dictated by a certain threshold, compute the TPR (true positive rate). From there, multiple statistics (such as AUC for the TPR-FPR curve, or TPR at a specific FPR) can be computed. I would request the authors to re-evaluate their methods using this approach. I also do not know why the authors have constructed their own version of baselines- there are existing attacks (as one example, see Min-k%++)

- Appendix E.1: "... have assumed complete knowledge of the original training questions" This is a very strong assumption! This implies that the adversary also knows the exact questions associated with the target document (and by extension the answers). This was not mentioned clearly anywhere in the main paper. The fact that performance drops when this is relaxed is not surprising to me, and demonstrates that the majority of performance reported in the paper is originating from this strong (and obfuscated) assumption. The relaxed setting here (where exact questions are not known) should be the **default** setting in the main paper- after all, true "document" inference would not entail one knowing which *exact* questions and answers were present in the training data


## Minor comments

- L31: "...fuels a significant number of operations daily" - source?
- L49-50: "...utilize a dual representation..." - please cite relevant works
- L102-102: "The literature indicates that...." - this is not the case. While recent work [1] does suggest that parameter access may be necessary for optimal membership inference, the cited papers here (and most empirical results) indeed conclude that additional access to parameters does not help with membership inference.
- L114: "...larger scale models"- please see [2] that indeed tackles inferring the use of certain text documents for LLMs.
- L186: "adversary lacks access to an auxiliary data". Membership-inference evaluation by design needs "non-members" to compute thresholds for membership classification (and as recent work [3] has shown, you cannot just use any non-member data especially when it comes to language). What the authors probably mean here is that there is not *enough* auxiliary data, and should clarify this.
- L191: "...would be prohibitively expensive" - this would be a problem for a resource-constrained adversary, but an adversary that is malicious (in the security sense) will not be limited like this. An adversary can always use public/pretrained models as a starting point.
- L210: "...types of documents" - define knowledge of "type" of documents/questions
- L447: "Table 1 (right)" - there is only one table. Please fix reference (similarly for Table 1 (left), etc.)

### References
- [1] Suri, Anshuman, et al. "Do Parameters Reveal More than Loss for Membership Inference?" HiLD, ICML (2024).
- [2] Maini, Pratyush, et al. "LLM Dataset Inference: Did you train on my dataset?" arXiv:2406.06443 (2024).
- [3] Duan, Michael, et al. "Do membership inference attacks work on large language models?" arXiv:2402.07841 (2024).

**Questions:**

- L106: why is the adversary query-restricted?

- Section 4.2.1: How is convergence defined here? There is some (unintentional) leakage happening, since you need to know when "convergence" happens for members to stop the optimization at that point. A true test to check for this would be to run it for a pre-defined set of iterations, record the loss at each iteration, and use that as the feature vector.

- L354: How are gradients back-propagated via the OCR generation pipeline?

- Section 5.3: How are the questions constructed? Are they picked based on what was in train/test data?

- L410: What is "predicted labels"? Isn't the task to generate an answer? If so, just say 'generated text'

- L1008: "...paraphrase the original training questions" - do their answers remain the same?

---

> ### Author Response · Authors · 2024-11-21
> **Answers to Reviewer  **V2Bv****
>
> We appreciate the reviewer’s insightful feedback.
>
> **W1: The evaluation design is completely disconnected from how MI attacks are usually evaluated. Currently, thresholds are computed based on average values in D_test. In MI (or similar evaluations), the standard protocol is to sweep over the non-member's values and based on the FPR (false positive rate) dictated by a certain threshold, compute the TPR (true positive rate). From there, multiple statistics (such as AUC for the TPR-FPR curve, or TPR at a specific FPR) can be computed. I would request the authors to re-evaluate their methods using this approach.**
>
> Regarding the evaluation protocol, we have re-evaluated the attack performance for all methods, focusing on TPR at 1% and 3% FPR. The updated results are reported in the Appendix E, where we observe that our proposed attacks demonstrate strong performance in comparison to the baselines.
>
> **W1: I also do not know why the authors have constructed their own version of baselines- there are existing attacks (as one example, see Min-k%++)**
>
> To the best of our knowledge, there are no existing document-level membership inference attacks specifically designed for DocVQA settings, where each document is associated with multiple questions and individual losses. Adapting existing attacks to this context could be considered a contribution in itself. Therefore, we focused on two main baselines that do not require any additional auxiliary datasets, as they do not rely on shadow datasets—aligning with the settings assumed for our proposed white-box and black-box attacks. Specifically, we adapted the provider-membership inference attack [8] for document-level inference, as it is inherently suited for DocVQA settings, and we included the well-known Yeom attack [1], which is widely accepted as a baseline in membership inference research [1,2,3,4,5] and has even been used as a baseline in Min-K%++.
>
> Additionally, regarding the suggested baseline, we note that a recent SoK paper [7] has highlighted significant limitations in its evaluation. Specifically, it was shown that datasets used in such evaluations suffer from distribution shifts between members and non-members. This allows simple classifiers, such as a bag-of-words model, to achieve high AUC values without leveraging any model-specific information, thus calling into question the validity of its results as a measure of model memorization. Therefore, while we will present these results, they should be interpreted in light of this critique. Furthermore, the paper introducing the suggested baseline is currently under review at ICLR and has not yet been accepted at any venue.
>
> We compare our proposed attacks with both Min-K% and Min-K%++. We make use of the publicly available code from the paper and adapt it for the DocMIA: (1) We use the average AVG to aggregate the Min-K%++ scores across all questions associated with each document, (2) As the predicted answers in DocVQA models is shorted compared to the generated text from LLMs, we experiment with K values in [0.6,0.7,0.8,0.9,1.0] and report the best-performing result. The TPR@3%FPR results are summarized in the table below (full results are provided in Appendix E), showing that our methods outperform Min-K%++ in DocMIA against most target models.
>
> |||DocVQA|||PFL&nbsp;&nbsp;&nbsp;&nbsp;&nbsp;&nbsp;&nbsp;&nbsp;&nbsp;&nbsp;&nbsp;&nbsp;|
> |------|:------:|:------:|:------:|:------:|:------:|
> | |VT5| Donut|Pix2Struct-B|VT5|Donut|
> |**Min-K%**|**10.67**|1.00|5.33|5.67|0.00|
> |**Min-K%++**|10.00| 9.33 |10.33|8.00|1.67|
> |**FL (Ours)**| 5.67 |**10.67**|**11.00**|**8.67**|**7.00**|
>
> [1]: Yeom, Samuel, et al. "Privacy risk in machine learning: Analyzing the connection to overfitting." 2018 IEEE 31st computer security foundations symposium (CSF). IEEE, 2018.
>
> [2]: Carlini, Nicholas, et al. "Membership inference attacks from first principles." 2022 IEEE Symposium on Security and Privacy (SP). IEEE, 2022.
>
> [3]: Zarifzadeh, Sajjad, Philippe Liu, and Reza Shokri. "Low-Cost High-Power Membership Inference Attacks." Forty-first International Conference on Machine Learning. 2024.
>
> [4]: Shi, Weijia, et al. "Detecting pretraining data from large language models." arXiv preprint arXiv:2310.16789 (2023). Published as a conference paper at ICLR 2024.
>
> [5]: Zhang, Jingyang, et al. "Min-k%++: Improved baseline for detecting pre-training data from large language models." arXiv preprint arXiv:2404.02936 (2024).
>
> [6]: Shi, Weijia, et al. "Detecting pretraining data from large language models." arXiv preprint arXiv:2310.16789 (2023). Published as a conference paper at ICLR 2024.
>
> [7]: Meeus, Matthieu, et al. "SoK: Membership Inference Attacks on LLMs are Rushing Nowhere (and How to Fix It)." arXiv preprint arXiv:2406.17975 (2024).
>
> [8]: Tito, Rubèn, et al. "Privacy-aware document visual question answering." International Conference on Document Analysis and Recognition. Cham: Springer Nature Switzerland, 2024.

---

> > ### Comment · Reviewer_V2Bv · 2024-11-21
> >
> > > The updated results are reported in the Appendix E, where we observe that our proposed attacks demonstrate strong performance in comparison to the baselines.
> >
> > While I appreciate the authors adding these results, my point was that this method of evaluation is the standard for MI evaluations. It should be the other experiments (if any) that should be in the Appendix, not the other way round. As far as performance under this metric goes- was any bootstrapping used for the values? If only 300 members and non-members are used, TPR at a low FPR like 1% will not be very stable. The gap for TPR @ 3% FPR is not impressive and is in fact in most datasets worse for the proposed method (Table 9). I also do not understand why Min-K% is listed under white-box attacks; the attack does not require model parameters.
> >
> > >  This allows simple classifiers, such as a bag-of-words model, to achieve high AUC values without leveraging any model-specific information, thus calling into question the validity of its results as a measure of model memorization
> >
> > I am aware of this work (and other related critiques) - their claims hold for classic text-level membership inference under LMs, but the setting here is different (which is the very motivation of the work) given how documents can be repeated and be rich in information. Using this work as justification to not use existing baselines does not make sense to me. If anything, including those attacks and then (potentially) confirming that they do not work for document-level inference either would be a useful contribution in itself.

---

> > > ### Author Response · Authors · 2024-11-23
> > > **Answers to Reviewer **V2Bv****
> > >
> > > **While I appreciate the authors adding these results, my point was that this method of evaluation is the standard for MI evaluations. It should be the other experiments (if any) that should be in the Appendix, not the other way round.**
> > >
> > > Thank you for your valuable feedback. As per your suggestion, we have now included the TPR@FPR metric in the main result tables of the paper to align with standard evaluation practices.
> > >
> > > **As far as performance under this metric goes- was any bootstrapping used for the values? If only 300 members and non-members are used, TPR at a low FPR like 1% will not be very stable.**
> > >
> > > Our proposed attacks utilize five random seeds for the KMeans algorithm to cluster member and non-member documents, and we report the average performance across these runs. We evaluate the attacks on a data split where documents are randomly sampled from the original train/test set. For these documents, the answers remain consistent, while the questions are either directly reused or paraphrased.
> > >
> > > Finally, we believe that our extensive experiments on four DocVQA models across two DocVQA datasets demonstrate that our proposed attacks consistently achieve strong performance and robustness against various target models. These results underscore the effectiveness of our methods in the DocMIA setting.
> > >
> > > **The gap for TPR @ 3% FPR is not impressive and is in fact in most datasets worse for the proposed method (Table 9).**
> > >
> > > Thank you for your thoughtful feedback. We would like to clarify that, among our multiple approaches in Table 11 (Table 9 in the previous version), the only significant gap where any of our methods perform worse compared to the best baseline is on the DocVQA dataset with VT5, specifically when considering TPR@1%FPR. In the majority of cases, however, our approaches outperform the baselines, and when they do not, the performance gap is minimal across each setting. Additionally, these improvements are further accentuated in Table 12 (Table 10 in the previous version), where we demonstrate results under black-box access, highlighting the robustness of our methods in more constrained environments.
> > >
> > > **I also do not understand why Min-K% is listed under white-box attacks; the attack does not require model parameters.**
> > >
> > > Thank you for your comment. We appreciate your observation regarding the classification of the Min-K% attack. To clarify, Min-K%[1] and Min-K%++[1] require grey-box access (as mentioned in [2]) to the target model, specifically the ability to access token probabilities. While this access level is less than full white-box access, it still provides more information than black-box access, where only the words (not even tokens) are available.
> > >
> > > Since our paper is focused on attacks under white-box and black-box settings, and considering that Min-K% and Min-K%++ fall between these two access levels, we believe it is more appropriate to classify them within the white-box category. However, we will update the text to better explain this distinction and ensure the classification is clear to readers.
> > >
> > > Thank you again for your helpful feedback!
> > >
> > > **I am aware of this work (and other related critiques) - their claims hold for classic text-level membership inference under LMs, but the setting here is different (which is the very motivation of the work) given how documents can be repeated and be rich in information. Using this work as justification to not use existing baselines does not make sense to me. If anything, including those attacks and then (potentially) confirming that they do not work for document-level inference either would be a useful contribution in itself.**
> > >
> > > We apologize if our comment was misinterpreted. We would like to clarify that in  our initial submission, we were unaware of the Min-K%[1] and Min-K%++[2] baselines. However, in our rebuttal, we have incorporated comparisons with these methods in our experiments and the results were added to the revised version. These results demonstrate that our proposed attacks outperform these baselines in the DocMIA setting across most target models and datasets.
> > >
> > > We are once again grateful that you suggested these baselines to compare with our approaches, as it will help strengthen our results.
> > >
> > > [1]: Shi, Weijia, et al. "Detecting pretraining data from large language models." arXiv preprint arXiv:2310.16789 (2023). Published as a conference paper at ICLR 2024.
> > >
> > > [2]: Zhang, Jingyang, et al. "Min-k%++: Improved baseline for detecting pre-training data from large language models." arXiv preprint arXiv:2404.02936 (2024).

---

> ### Author Response · Authors · 2024-11-21
> **Answers to Reviewer **V2Bv****
>
> **W2: Appendix E.1: "... have assumed complete knowledge of the original training questions" This is a very strong assumption! This implies that the adversary also knows the exact questions associated with the target document (and by extension the answers). This was not mentioned clearly anywhere in the main paper. The fact that performance drops when this is relaxed is not surprising to me, and demonstrates that the majority of performance reported in the paper is originating from this strong (and obfuscated) assumption. The relaxed setting here (where exact questions are not known) should be the default setting in the main paper- after all, true "document" inference would not entail one knowing which exact questions and answers were present in the training data**
>
> We would like to thank the reviewer for their valuable feedback. We apologize for any confusion regarding the assumption of complete knowledge of the original training questions. This assumption was included to evaluate upper-bound performance, and we did not intend to obfuscate our results. Initially, we planned to include the results from Appendix E.1 (Appendix F.2 in the revised version of the paper) in the main body of the paper, but due to space constraints, we moved them to the appendix.
>
>
> We fully agree with the reviewer that, in realistic scenarios, an adversary is unlikely to have access to the exact training questions. However, we argue that an adversary with domain knowledge can approximate training questions based on the type of documents used for training. For instance, in the DocVQA setting, the type of questions is often closely tied to the document type. For example, if the model is trained on invoices, natural questions might include "What is the total?" to extract the total amount. Thus, an adversary could craft questions that are similar in distribution to the original training questions. Moreover, we believe it is important to consider cases where part of the preprocessed dataset may be leaked, providing the adversary with question-answer pairs in the format (question, answer, document).
>
> While manual crafting of such questions and documents is a more natural approach for constructing evaluation sets, this process is slow for scaling up. In this work, we opted for a more efficient method by using LLMs to rephrase the original questions, enabling large-scale evaluation. We consider this as part of the experimental setup, despite it not representing an optimal adversary's approach to obtaining questions, which thus diverges from their natural form.
>
> Moreover, we have shown in Appendix E.1 (Appendix F.2 in the revised version of the paper) that even when exact training questions are unavailable and rephrased questions generated using LLMs are used, we still demonstrate stronger performance than the baselines. This highlights the robustness of our approaches in more realistic scenarios.
>
> We updated the threat model section to clearly acknowledge the assumption of complete knowledge of the original training questions, and we referenced Appendix E.1 (Appendix F.2 in the revised version of the paper) for additional results in the case where the questions are not known. We hope that this clarification addresses the concerns raised by the reviewer.
>
> **L31: "...fuels a significant number of operations daily" - source?**
>
> - The Global Document Management System Market size is expected to reach $11.3 billion by 2029, rising at a market growth of 10.7% CAGR during the forecast period ([ReportLinker](https://www.globenewswire.com/news-release/2023/06/20/2691459/0/en/The-Global-Document-Management-System-Market-size-is-expected-to-reach-11-3-billion-by-2029-rising-at-a-market-growth-of-10-7-CAGR-during-the-forecast-period.html), June 2023)
> - The global intelligent document processing market size was estimated at USD 2.30 billion in 2024 and is projected to grow at a CAGR of 33.1% from 2025 to 2030. ([GVR](https://www.grandviewresearch.com/industry-analysis/intelligent-document-processing-market-report), 2024)
> - There are over 4 trillion paper documents in the U.S. alone and they are growing at a rate of 22% per year (PricewaterhouseCoopers)
> - Just in invoice processing, most businesses in the US handle up to 500 invoices each month ([Global Report: Accounts Payable Automation Trends](https://www.webexpenses.com/us/resource/global-report-accounts-payable-automation-trends/))

---

> > ### Comment · Reviewer_V2Bv · 2024-11-21
> >
> > > Moreover, we have shown in Appendix E.1 (Appendix F.2 in the revised version of the paper) that even when exact training questions are unavailable and rephrased questions generated using LLMs are used, we still demonstrate stronger performance than the baselines. This highlights the robustness of our approaches in more realistic scenarios.
> >
> > While the example question provided about the invoice total is not very "unique", do the datasets tested in this paper comprise of such straightforward questions? Even if a paraphrased version is used, there is potential for leakage through the paraphrase itself. Perhaps the authors can consider automating question generation by providing the document to another LLM and asking it to generate questions?
> >
> > > We updated the threat model section to clearly acknowledge the assumption of complete knowledge of the original training questions, and we referenced Appendix E.1 (Appendix F.2 in the revised version of the paper) for additional results in the case where the questions are not known. We hope that this clarification addresses the concerns raised by the reviewer.
> >
> > Good- thank you

---

> > > ### Author Response · Authors · 2024-11-23
> > > **Answers to Reviewer **V2Bv****
> > >
> > > **While the example question provided about the invoice total is not very "unique", do the datasets tested in this paper comprise of such straightforward questions?**
> > >
> > > Yes, the datasets include questions that are generally straightforward and designed to extract key information.
> > >
> > > In DocVQA [1], the questions/answers are annotated by humans followed by a verification process to ensure that the questions are clear and unambiguous.
> > >
> > > In PFL-DocVQA [2], the questions/answers are based on a set of human-defined templates specifically designed to extract key information from invoices.
> > >
> > > Below are examples of actual questions from these datasets:
> > > DocVQA: 'What is the Venue Owner Name?', 'Specify the period starting date?', 'What type of report is this?'
> > > PFL: 'What is the mentioned estimate number?', 'What is the specific tax amount?', 'What is the individual net total amount?'
> > >
> > > [1] Matthew et al. "DocVQA: A Dataset for VQA on Document Images", WACV (2021).
> > >
> > > [2] Tito, Rubèn, et al. "Privacy-aware document visual question answering." International Conference on Document Analysis and Recognition. Cham: Springer Nature Switzerland, 2024.
> > >
> > > **Even if a paraphrased version is used, there is potential for leakage through the paraphrase itself. Perhaps the authors can consider automating question generation by providing the document to another LLM and asking it to generate questions?**
> > >
> > > Thank you for your valuable feedback. We agree that while the paraphrasing approach may leak less information compared to the approach where we assume prior knowledge of the training questions, it could still potentially introduce privacy leaks. However, we would like to emphasize that, in both scenarios (whether we use knowledge of the training questions or paraphrase them), all approaches, including the baselines, have access to the same level of information by using either the training questions or their paraphrases. We ensure that no method is privileged by providing more information, as we aim to maintain fairness in our evaluation.
> > >
> > > We also appreciate your suggestion of using an LLM to generate questions, which would indeed address the potential leakage issue. However, this solution would be time-consuming and would require careful control of each question-answer pair. In fact, we believe that publishing such a benchmark dataset could be a valuable contribution in itself and would be an interesting direction for future work.
> > >
> > > Finally, we would like to highlight that the suggested Min-K%++ paper considers the same two settings as we did with the questions: the original and the paraphrased. The former assesses the detection of verbatim training texts, while the latter paraphrases the training texts (using ChatGPT) and evaluates them on paraphrased inputs.
> > >
> > > Thank you again for your constructive suggestion!

---

> ### Author Response · Authors · 2024-11-21
> **Answers to Reviewer **V2Bv****
>
> **L102-102: "The literature indicates that...." - this is not the case...**
>
> In [1], the authors propose two strategies, abbreviated as $\mathcal{I}_A$ and $\mathcal{I}_V$ (please refer to Sections 3.3 and 3.4 in [1]), which require running adversarial attacks, such as the Projected Gradient Descent (PGD) method. Consequently, these strategies necessitate white-box access to the model. The results (Tables 2 to 11) for these two strategies demonstrate that they often outperform the approach requiring only black-box access, abbreviated as $\mathcal{I}_B$.
>
> In [2], it is demonstrated that using the outputs of the model's activation functions as attack features in white-box settings does not yield better performance compared to using features accessible only in black-box settings. However, when more informative features, such as the gradient norm and the loss, are selected in the white-box setting, the resulting attack performance surpasses that of the black-box setting (please refer to Table VIII in [2]).
>
> [1]: Song, Liwei, Reza Shokri, and Prateek Mittal. "Privacy risks of securing machine learning models against adversarial examples." Proceedings of the 2019 ACM SIGSAC conference on computer and communications security. 2019.
>
> [2]: Nasr, Milad, Reza Shokri, and Amir Houmansadr. "Comprehensive privacy analysis of deep learning: Passive and active white-box inference attacks against centralized and federated learning." 2019 IEEE symposium on security and privacy (SP). IEEE, 2019.
>
> **L114: "...larger scale models"- please see [2] that indeed tackles inferring the use of certain text documents for LLMs.**
>
> We thank the reviewer for mentioning this paper, which investigated the use of MIA scores to determine whether a dataset (comprising multiple data points) was used in the training process of large language models. While their attack focuses on text documents, it is important to emphasize that the setting differs significantly from DocVQA, where we compute a loss for each piece of information within a document. Furthermore, dataset inference attacks, represent a different class of attacks, somewhat analogous to the provider inference attack presented in [1], but their goals are less fine-grained than document-level MIA attacks. Finally, unlike our approach, which does not rely on additional data, their method requires auxiliary data to generate features for training the attack model.
>
> [1]: Tito, Rubèn, et al. "Privacy-aware document visual question answering." International Conference on Document Analysis and Recognition. Cham: Springer Nature Switzerland, 2024.
>
> **L186: "adversary lacks access to an auxiliary data". Membership-inference evaluation by design needs "non-members" to compute thresholds for membership classification...**
>
> Thank you for your valuable feedback and for pointing out this clarification. Indeed, by "auxiliary data," we meant that the adversary does not possess sufficient data to generate shadow datasets for training shadow models. These shadow models are typically used to extract features for training an attack model. We will revise the text to make this distinction clearer and align it with your suggestion.
>
> **L191: "...would be prohibitively expensive" - this would be a problem for a resource-constrained adversary, but an adversary that is malicious (in the security sense) will not be limited like this...**
>
> Thank you for your insightful feedback. While it's true that malicious adversaries may leverage public or pretrained models as a starting point, our paper specifically addresses the practical challenges involved in training shadow models on domain-specific documents. Even when using pretrained models, fine-tuning them to replicate the target model's behavior in a domain-specific context demands considerable computational resources. As such, the financial and computational burden of training shadow models on specialized data remains a significant obstacle [1,2], even for a determined adversary. This challenge is particularly pronounced when working with large-scale document datasets, where the cost of training robust shadow models with numerous parameters can quickly become prohibitively expensive [2].
>
> [1]: Carlini, Nicholas, et al. "Membership inference attacks from first principles." 2022 IEEE Symposium on Security and Privacy (SP). IEEE, 2022.
>
> [2]: Meeus, Matthieu, et al. "SoK: Membership Inference Attacks on LLMs are Rushing Nowhere (and How to Fix It)." arXiv preprint arXiv:2406.17975 (2024).

---

> ### Author Response · Authors · 2024-11-21
> **Answers to Reviewer **V2Bv****
>
> **L210: "...types of documents" - define knowledge of "type" of documents/questions**
>
> The "types of documents/questions" refers to the natural way of querying key information contained within a document. This aligns with the typical question-answer annotations defined by human annotators/experts for DocVQA datasets like TAT-QA[1] (Section 2.2); DocVQA[2] (Section 3.1), designed to effectively train practically useful DocVQA models. The type of question is inherently linked to the type of document on which the DocVQA model is trained. For instance, if the target model is trained on invoices, the natural type of question would focus on extracting essential details from the invoice, such as the “total amount”, framed in a clear and straightforward manner e.g., "What is the total?". We will clarify this point in the main paper.
>
> [1] Matthew et al. "DocVQA: A Dataset for VQA on Document Images", WACV (2021).
>
> [2] Zhu et al. "TAT-QA: A Question Answering Benchmark on a Hybrid of Tabular and Textual Content in Finance", ACL (2021).
>
> **Q1: why is the adversary query-restricted?**
>
> A simple privacy ad-hoc defense in black-box settings could be to limit the number of queries between the user and the target model.
>
> **Q2: Section 4.2.1: How is convergence defined here?...**
>
> In Section 4.1.2, convergence is defined as the point where the optimization process is stopped either by reaching the maximum number of iterations S or when the change in loss value falls below a threshold τ, as outlined in Algorithm 1. Importantly, in all our experiments, these hyperparameters S and τ are fixed for both members and non-members, ensuring no unintentional leakage.
>
> "The reviewer suggests using a fixed number of iterations and recording the loss at each iteration as a feature vector".
> We initially experimented with this approach by fixing the maximum number of iterations without applying a loss-change threshold. However, we found that this introduced significant noise into the feature vector, particularly due to the excessive number of optimization steps. As a result, this approach yielded inferior performance compared to our proposed method.
>
> **Q3: L354: How are gradients back-propagated via the OCR generation pipeline?**
>
> We do not back-propagate gradients through the OCR generation pipeline. Instead, we compute gradients with respect to the document pixel values and use them for updates directly. We will clarify this point in the main paper.
>
> **Q4: Section 5.3: How are the questions constructed?...**
>
> We use the questions provided directly within the datasets. Each document in the dataset comes paired with a predefined set of questions along with corresponding answers.
>
> **Q6: L1008: "...paraphrase the original training questions" - do their answers remain the same?**
>
> Yes, the answers remain the same because they are derived from the original document, which is fixed and unchanged.

---

> > ### Comment · Reviewer_V2Bv · 2024-11-22
> >
> > > ...reaching the maximum number of iterations S or when the change in loss value falls below a threshold τ, as outlined in Algorithm 1
> >
> > I hope that the authors can see my concern here. This threshold is tuned based on results observed, which require knowledge of members and non-members. Are the same hyper-parameters used across all models and datasets? If not, this is a considerable source of information leakage.
> >
> > > Yes, the answers remain the same because they are derived from the original document, which is fixed and unchanged.
> >
> > Is there only one way to write the given answer? If not, using the exact same answers means the adversary also knows the answer in its exact form, which is another assumption on top of the queries being exactly the same.

---

> ### Author Response · Authors · 2024-11-23
>
> **I hope that the authors can see my concern here. This threshold is tuned based on results observed, which require knowledge of members and non-members. Are the same hyper-parameters used across all models and datasets? If not, this is a considerable source of information leakage.**
>
> We sincerely thank the reviewer for their insightful comment. As outlined in Section D (Section C in the previous version), we performed a grid search for each of the three proposed methods. Our analysis indicated that the hyperparameters are model-dependent rather than dataset-dependent. To address your question, we confirm that the same hyperparameters are used for each approach across all datasets. The best parameters selected during our tuning process are shown in Table 7 (Table 5 in the previous version), where we highlight that these are the best hyperparameters from our tuning process, demonstrating consistent performance across both the PFL and DocVQA datasets.
>
> **Is there only one way to write the given answer? If not, using the exact same answers means the adversary also knows the answer in its exact form, which is another assumption on top of the queries being exactly the same.**
>
> In the DocVQA task, the goal is for the model to extract the exact content requested by the question directly from the document. We are not expecting the model to reformulate or paraphrase the answer. Therefore, there is only one correct way to write the given answer, as it corresponds to the precise information found in the document. This property is fundamental to the task, as we aim to evaluate the model's ability to correctly extract and match the content rather than produce variations of the answer.
>
> **Note**: We have mistakenly reported incorrect values in our evaluation of Min-K/Min-K++ in Table 11 (Table 9 in the previous version). We are sorry for any inconvenience. We have corrected and verified the results of both methods in the new version that we just uploaded.

---

> > ### Comment · Reviewer_V2Bv · 2024-11-23
> >
> > Thank you to the authors for all the additional experiments and revisions made during the rebuttal period. While it is unlikely that other attacks in MI evaluations for LLMs would work any better, I would encourage adding a few more baselines. Perhaps the recent EMNLP best paper (https://arxiv.org/abs/2409.14781) would be a good baseline to include. This is of course not a requirement for the rebuttal process, but would be nice to have in a later version.
> >
> > > We have mistakenly reported incorrect values in our evaluation of Min-K/Min-K++ in Table 11 (Table 9 in the previous version). We are sorry for any inconvenience.
> >
> > I hope the reviewers are confident in their evaluations and current results after whatever fix has been applied. Including an anonymized link to the codebase would be very helpful (I hope the authors will add it by the final version).
> >
> > As an unrelated minor comment, the aspect ratio of Figure 2 seems off.
> >
> > Seeing that most of my concerns have been addressed, I will be increasing my score to 6. The reason it is not higher is: while the proposed methods work well, not any one of them works consistently well. For instance looking at Table 2, FL seems to work best in most cases but where it is not the best-performing, it is at least 2x worse than existing baselines. I would like to know what the authors think of this.

---

> ### Author Response · Authors · 2024-11-23
>
> We sincerely thank the reviewer for their positive feedback and for recognizing the efforts made during the rebuttal process.
>
> **...I would encourage adding a few more baselines. Perhaps the recent EMNLP best paper (https://arxiv.org/abs/2409.14781) would be a good baseline to include...**
>
> We greatly appreciate the suggestion to include the EMNLP best paper as a baseline for membership inference evaluations on LLMs. While it was not possible to incorporate this within the rebuttal period, we will do our best to include the results of this baseline in the camera-ready version of our paper.
>
> **I hope the reviewers are confident in their evaluations and current results after whatever fix has been applied. Including an anonymized link to the codebase would be very helpful (I hope the authors will add it by the final version).**
>
> We would like to clarify that the error in the previously reported values for Min-K/Min-K++ in Table 11 (Table 9 in the earlier version) was not due to a bug in the code but occurred only during the reporting of results for these two new baselines (and only for these two new baselines). We have carefully corrected this mistake and double-checked the results. We are pleased to confirm that the reported results are accurate for all baselines and all our approaches.
>
> Regarding the code, as mentioned in Section 9, we aim to release it for the camera-ready version of the paper. We are currently enhancing the code by adding comments to make it more user-friendly for readers and researchers.
>
> **...while the proposed methods work well, not any one of them works consistently well. For instance looking at Table 2, FL seems to work best in most cases but where it is not the best-performing, it is at least 2x worse than existing baselines...**
>
> Regarding your observation about the performance consistency of the proposed methods: We agree that while FL performs best in most cases, its performance does drop significantly in certain scenarios compared to existing baselines. (1) The variation in performance is primarily influenced by the complexity of the task and the inherent sensitivity of different models or settings to specific attack strategies. We believe this highlights the importance of having a diverse set of approaches, as no single method can uniformly outperform across all scenarios. (2) Particularly, we observe that the LOSS-TA baseline performs best against Donut on PFL and VT5 on DocVQA. As discussed in Appendix F (White-box Results), this performance can be attributed to the significant gap in average loss values between member and non-member samples (for instance, see Figure 7, top left in case of VT5 on DocVQA.). Such a gap strongly indicates overfitting in these cases, enabling the LOSS-TA method to effectively exploit this characteristic.
>
> In future work, we aim to investigate ways to enhance the robustness of FL and other methods, potentially by adapting them dynamically to specific scenarios or combining strengths from multiple approaches.
>
> Thank you again for your valuable insights and for encouraging us to think further about this important aspect of our work.

---

### Official Review · Reviewer_7wUB · 2024-10-31

**Soundness:** 3
**Presentation:** 2
**Contribution:** 2
**Rating:** 6
**Confidence:** 2

**Summary:**

This paper introduce the first document-level membership inference attacks Document-Level Membership Inference Attacks (DocMIA) for Document Visual Question Answering (DocVQA) models, addressing privacy risks in multimodel contexts. Through extensive experiments across multiple datasets and models, significantly outperform existing membership inference baseline.

**Strengths:**

1、This paper proposes Document-Level Membership Inference Attacks (DocMIA), which deal with the multiple appearances of the same document in the training set.

2、DocMIA consistently achieves high performance across all target models, the strong performance of  DocMIA highlights the privacy risks posed by optimization-based features of membership.

**Weaknesses:**

1、The writing in this paper is clear and accessible, providing a well-structured and understandable presentation of the authors' ideas and methods. However, the text is overly verbose, with an excess of written content and a lack of visual aids. The absence of a diagram illustrating the attack process detracts from an intuitive understanding of the attack methodology. Adding visual elements would enhance the reader's comprehension and provide a more direct insight into the core processes discussed.

2、There are some typographical problems with the manuscript. The position of Figure 1 is not ideal and should be adjusted to be centered side-by-side, similar to Figure 2.

**Questions:**

1、The paper presents experiments on four target models for the DocVQA dataset but only conducts experiments on two models for the PEL-DocVQA dataset, without providing an explanation for this choice of setup. This lack of clarification on the experimental design for the PEL-DocVQA dataset leaves the rationale for the selection of models unclear. It is recommended to include an explanation for this experimental setting.

---

> ### Author Response · Authors · 2024-11-21
> **Answers to Reviewer  **7wUB****
>
> **W1: The writing in this paper is clear and accessible, providing a well-structured and understandable presentation of the authors' ideas and methods. However, the text is overly verbose, with an excess of written content and a lack of visual aids. The absence of a diagram illustrating the attack process detracts from an intuitive understanding of the attack methodology. Adding visual elements would enhance the reader's comprehension and provide a more direct insight into the core processes discussed. + W2: There are some typographical problems with the manuscript. The position of Figure 1 is not ideal and should be adjusted to be centered side-by-side, similar to Figure 2.**
>
> Thank you for your valuable feedback. We agree that a diagram illustrating the attack process would greatly enhance the reader's understanding. While we have included Figures 4 and 5 in the Appendix to provide a detailed visual representation of the attack methodology, space limitations prevented us from including them in the main part of the paper.
>
> **Q1: The paper presents experiments on four target models for the DocVQA dataset but only conducts experiments on two models for the PFL-DocVQA dataset, without providing an explanation for this choice of setup. This lack of clarification on the experimental design for the PFL-DocVQA dataset leaves the rationale for the selection of models unclear. It is recommended to include an explanation for this experimental setting.**
>
> In this work, we prioritize the use of publicly available checkpoints for all target models whenever possible. This decision ensures the reproducibility of our results and minimizes potential biases that could arise from self-trained models. For example, models trained on our limited computational resources might not match the performance reported by their respective authors due to differences in training setups and resource constraints (as noted for Donut and Pix2Struct in [1][2]).
>
> As mentioned in the Appendix D.1, we consider only two models (VT5 and Donut) for the PFL-DocVQA dataset because public checkpoints for Pix2Struct are unavailable for this dataset. This lack of available checkpoints prevents us from including Pix2Struct as a target model for PFL-DocVQA. Indeed, we attempted to train Pix2Struct on PFL-DocVQA but did not succeed to achieve good performance.
>
> In contrast, for the DocVQA dataset, we utilize publicly available checkpoints for all considered target models (VT5, Donut, Pix2Struct-Base, and Pix2Struct-Large) provided by their respective authors. Unlike the PFL-DocVQA dataset, DocVQA is a well-established benchmark, making such resources more readily accessible.
>
> These considerations explain our choice of experimental setup and the inclusion of only two models for PFL-DocVQA.
>
> [1] Kim et al. "OCR-free Document Understanding Transformer", ECCV (2022).
>
> [2] Lee et al. "Pix2Struct: Screenshot Parsing as Pretraining for Visual Language Understanding", ICML (2023).

---

### Official Review · Reviewer_3nRY · 2024-11-03

**Soundness:** 3
**Presentation:** 2
**Contribution:** 3
**Rating:** 6
**Confidence:** 3

**Summary:**

The paper aims to make a fine-grained membership inference attach, which can determine whether a sensitive document is included in the training dataset. For the white box and black box, the authors have developed different MIA approaches according to their characteristics. Experimental results confirm that their methods outperform baselines.

**Strengths:**

1. This paper is the first attempt to target document-level membership inference attacks.
2. This paper designs attack modes for white box and black box respectively.

**Weaknesses:**

1. The challenges of the work are underdescribed by the authors, and it is difficult for the reviewer to understand the technical challenges between fine-grained inference attacks and coarse-grained attacks, proposed by Tito et al.. The authors express it more as if they were writing an experimental report.
2. "shadow training of proxy models becomes infeasible.", the reviewer suggests the authors describe in words how to solve this problem and highlight it.
3. While the paper introduces methods to adapt the attack to black-box models through knowledge transfer, this approach may not be as effective as direct white-box attacks due to the inherent limitations in approximating the behavior of complex models.

**Questions:**

1. Briefly describe how to solve “shadow training of proxy models becomes infeasible".
2. While the paper introduces methods to adapt the attack to black-box models through knowledge transfer, this approach may not be as effective as direct white-box attacks due to the inherent limitations in approximating the behavior of complex models.

---

> ### Author Response · Authors · 2024-11-21
> **Answers to Reviewer  **3nRY****
>
> **W1: The challenges of the work are underdescribed by the authors, and it is difficult for the reviewer to understand the technical challenges between fine-grained inference attacks and coarse-grained attacks, proposed by Tito et al.. The authors express it more as if they were writing an experimental report.**
>
> Thank you for your insightful comment. We apologize if the distinction between fine-grained and coarse-grained inference attacks was not sufficiently clear. As mentioned in both the introduction and the related work section, the main difference lies in the inference goal: while Tito et al.'s work aims to infer whether a provider, which generates multiple documents (some of which may be part of the training dataset), was included in the training process, our focus is on determining the membership information of a single document. In our approach, we rely solely on one document to infer its membership status, whereas Tito et al.'s method can leverage multiple documents from the same provider. Additionally, we adapted their approach to document-level inference and considered it as a baseline for our work. We appreciate your feedback and will revise the related work section to clarify this difference further.
>
> **W2: "shadow training of proxy models becomes infeasible.", the reviewer suggests the authors describe in words how to solve this problem and highlight it. + Q1: Briefly describe how to solve “shadow training of proxy models becomes infeasible".**
>
> To address the challenge of shadow training becoming infeasible, we propose attacks that eliminate the reliance on auxiliary datasets and the need to train supervised models on features generated via shadow training. Instead, unsupervised attacks, such as clustering-based methods, provide a viable alternative. These approaches enhance efficiency by bypassing the shadow training step while maintaining strong attack performance, making them practical and scalable solutions for scenarios where shadow training is not feasible.
>
> **W3+Q2: While the paper introduces methods to adapt the attack to black-box models through knowledge transfer, this approach may not be as effective as direct white-box attacks due to the inherent limitations in approximating the behavior of complex models.**
>
> We would like to thank the reviewer for this observation. Indeed, we agree that the performance of our first approach in white-box settings is superior to that in black-box settings, which was expected. However, it is important to note that access to the target model is not always available in white-box settings. Therefore, it is crucial to demonstrate that even when restrictions are imposed to limit access to the model, private information can still be inferred, particularly in this case, the document-membership information.

---

### Official Review · Reviewer_hZiB · 2024-11-04

**Soundness:** 3
**Presentation:** 3
**Contribution:** 3
**Rating:** 6
**Confidence:** 3

**Summary:**

This paper introduces Document-level Membership Inference Attacks (DocMIA), focusing on Document Visual Question Answering (DocVQA) models. The paper presents two novel membership inference attacks tailored for DocVQA, targeting both white-box and black-box attack scenarios. These attacks enable adversaries to determine whether a particular document was part of the training set, even in the absence of auxiliary datasets. The proposed attacks leverage unsupervised optimization-based methods that outperform existing state-of-the-art membership inference attacks, underscoring the privacy vulnerabilities present in DocVQA models.

**Strengths:**

- The focus on DocVQA models for membership inference is unique. Previous research has mainly centered on generic models or other multimodal domains, while this paper contributes significantly to understanding privacy risks specific to document processing and VQA models.
- The paper employs solid, theoretically grounded approaches to devise the DocMIA attacks. The white-box attack uses optimization-based discriminative features, allowing more robust membership inference without requiring shadow models. The experimental setup includes multiple DocVQA models (VT5, Donut, Pix2Struct), ensuring that results are thorough and comparable.
- Given the sensitive nature of document data, this paper’s contributions are valuable for highlighting vulnerabilities in DocVQA models. The presented attacks have significant implications for the adoption of such models in privacy-sensitive applications, urging the development of more robust defenses.

**Weaknesses:**

-  While the paper references relevant literature, it does not include direct empirical comparisons with existing black-box defenses. Including black-box defenses as baselines would allow a more comprehensive assessment of DocMIA’s performance, helping to contextualize its strengths better.
- The experimental setup lacks some critical details, such as the specific training dataset used for GAN priors in particular baselines. Information on hyperparameters for certain attack baselines (e.g., gradient-based attacks) would improve reproducibility.
- The paper does not address the performance of DocMIA under dataset distribution shifts. This gap is significant as DocVQA models may be deployed across varied domains, potentially affecting the reliability of the proposed attacks. Testing DocMIA under different dataset shifts would provide insights into its robustness.

**Questions:**

How does DocMIA perform when there is a significant distribution shift between the public and private datasets? For instance, if the private dataset is derived from a financial dataset, but the public dataset stems from a different domain (e.g., legal documents), how would that affect the attack’s efficacy?

---

> ### Author Response · Authors · 2024-11-21
> **Answers to Reviewer  **hZiB****
>
> **W1: While the paper references relevant literature, it does not include direct empirical comparisons with existing black-box defenses. Including black-box defenses as baselines would allow a more comprehensive assessment of DocMIA’s performance, helping to contextualize its strengths better.**
>
> We thank the reviewer for the suggestion. We have evaluated the differentially private defense discussed in Appendix G, which provides theoretical privacy guarantees. We assess our attack on different privacy budgets and also provide the performance of the differentially private model. Please find the results in our response to Reviewer **HkZx**, which are also reported in Appendix G.
>
> **W2: The experimental setup lacks some critical details, such as the specific training dataset used for GAN priors in particular baselines. Information on hyperparameters for certain attack baselines (e.g., gradient-based attacks) would improve reproducibility.**
>
> We thank the reviewer for their remark. In Section 5, we have included descriptions of the models, dataset, and settings; Appendix C provides a detailed analysis of the hyperparameter tuning process for DocMIA, and Appendix D covers the implementation details of the attack. Additionally, all our models and datasets are publicly available, and as noted in Section 9, we are prepared to release the code for the camera-ready version. We have made considerable efforts to enhance the reproducibility and usability of our work. We are very grateful for any further suggestions, especially regarding hyperparameter details that may help make our work even more reproducible.
>
> **W3: The paper does not address the performance of DocMIA under dataset distribution shifts. This gap is significant as DocVQA models may be deployed across varied domains, potentially affecting the reliability of the proposed attacks. Testing DocMIA under different dataset shifts would provide insights into its robustness. + How does DocMIA perform when there is a significant distribution shift between the public and private datasets? For instance, if the private dataset is derived from a financial dataset, but the public dataset stems from a different domain (e.g., legal documents), how would that affect the attack’s efficacy?**
>
> We thank the authors for their thoughtful comments. We believe that 'public dataset' in this comment refers to the test dataset. If that's the case, first, the purpose of a membership inference attack is to infer whether a document was used in the model's training. Therefore, during evaluation, we should at least include documents used during training (members).
>
> Regarding non-included documents, we could indeed select documents with a significant distribution shift compared to members. However, this poses the risk of making the evaluation relatively easy and largely meaningless, as, instead of inferring membership information, a simple attack would aim merely to distinguish between the distributions, achieving an almost perfect score in the membership inference evaluation. In fact, it was recently shown in [1] that for the WikiMIA dataset, a membership inference evaluation dataset where members are Wikipedia snippets from before 2017 and non-members from after 2023, state-of-the-art results in membership inference were achieved by extracting simple features (such as dates and n-grams) and simply distinguishing between members and non-members.
>
> Thus, it is more challenging to infer membership information when members and non-members come from the same distribution [1,2,3,4]. Finally, it is generally realistic to assume that we know the distribution of data on which the models were fine-tuned, as these models are typically specialized for a particular data distribution to deliver the best results on that distribution.
>
> [1]: Das, Debeshee, Jie Zhang, and Florian Tramèr. "Blind baselines beat membership inference attacks for foundation models." arXiv preprint arXiv:2406.16201 (2024).
>
> [2]: Duan, Michael, et al. "Do membership inference attacks work on large language models?." arXiv preprint arXiv:2402.07841 (2024).
>
> [3]: Maini, Pratyush, et al. "LLM Dataset Inference: Did you train on my dataset?." arXiv preprint arXiv:2406.06443 (2024).
>
> [4]: Meeus, Matthieu, et al. "SoK: Membership Inference Attacks on LLMs are Rushing Nowhere (and How to Fix It)." arXiv preprint arXiv:2406.17975 (2024).

---

> > ### Comment · Reviewer_hZiB · 2024-11-29
> >
> > Thank you to the authors for taking the time to address my concerns.

---

> > > ### Author Response · Authors · 2024-11-29
> > >
> > > Thank you for your constructive feedback and for the raised score. We sincerely appreciate your time and efforts in reviewing our work.

---

### Official Review · Reviewer_i62x · 2024-11-04

**Soundness:** 3
**Presentation:** 3
**Contribution:** 3
**Rating:** 6
**Confidence:** 3

**Summary:**

This paper studies the document-level membership inference attacks targeting multi-modal models for document visual question answering. The main idea is to extract discriminate features between member and non-member documents based on their optimization trajectory. For black-box settings, a proxy model is utilized to extract discriminate features that are relevant for the black-box target model. Empirical results show that the proposed approach consistently performs well in various settings.

**Strengths:**

1. The idea of leveraging optimization trajectory for membership inference is novel.
2. The empirical performance of the proposed method is stable, indicating a good generalization over different attack settings.

**Weaknesses:**

1. There is a lack of ablation study on the choice of the features. In particular, giving some insights about what features are most useful for learning discriminative features between member and non-member documents. Of course, a conclusion of all obtained features being equally important or specific features being more relevant for certain applications are also good observations.
2. There is no clear explanation on the asymmetric transferability between different proxy and target model pairs Table 2. I think these are all key observations that demystify the success of membership inference attacks.

**Questions:**

Having additional ablation studies as well as providing more insights on the asymmetric transferability between different proxy and target models will improve the paper.

---

> ### Author Response · Authors · 2024-11-21
> **Answers to Reviewer **i62x****
>
> **W1: There is a lack of ablation study on the choice of the features. In particular, giving some insights about what features are most useful for learning discriminative features between member and non-member documents. Of course, a conclusion of all obtained features being equally important or specific features being more relevant for certain applications are also good observations. + Q1: Having additional ablation studies.**
>
> We provide a detailed analysis of the impact of individual features and their combinations on attack performance across all considered target models. Table below shows the impact of each feature in the case of Donut on DocVQA dataset. We refer the reader to Appendix F.1 for the complete results. Our findings indicate that the proposed optimization-based features consistently provide the most discriminative membership signals. Specifically, these features outperform other individual metrics, such as the DocVQA score, loss value, and gradient norm, when using the AVG aggregation function in most cases.
>
> | AVG(NLS) | AVG($\Delta$) | AVG($s$) | &nbsp; &nbsp;F1 |
> | :------: | :------: | :------: | :------: |
> |✓| | | 67.58 |
> | | ✓ | | 71.36 |
> | | |✓ | 73.16|
> |✓| ✓| | 72.87 |
> |✓| | ✓| 73.67 |
> |✓| ✓| | 73.86 |
> |✓| ✓| ✓| **73.89** |
>
> Additionally, incorporating multiple aggregation functions leads to additive improvements in attack performance, further demonstrating the robustness of our feature set. These results suggest that our proposed set of features is reliable and effective across various attack settings and target models.
>
> **W2:  There is no clear explanation on the asymmetric transferability between different proxy and target model pairs Table 2. I think these are all key observations that demystify the success of membership inference attacks. + Q2: Providing more insights on the asymmetric transferability between different proxy and target models will improve the paper.**
>
> We thank the reviewer for this insightful comment. Our intuition is that membership information is encoded in the outputs of the target (black-box) model, which serves as a teacher for the proxy model. During training, the proxy model parrots the behavior of the teacher model and consequently also learns the membership information. This phenomenon has been observed previously in [1], which supports our hypothesis.
>
> We add this intuition to Section 4.3 (Black-box DocMIA) of the paper to clarify our reasoning behind the observed asymmetry in transferability between proxy and target models.
>
> [1]: Jagielski, Matthew, et al. "Students parrot their teachers: Membership inference on model distillation." Advances in Neural Information Processing Systems 36 (2024).

---

> > ### Comment · Reviewer_i62x · 2024-11-25
> >
> > Thanks for the rebuttal and my concerns are now well-resolved. I have updated my score accordingly!

---

> > > ### Author Response · Authors · 2024-11-25
> > >
> > > Thank you for considering our rebuttal and for updating your evaluation. We truly appreciate your constructive feedback and are glad we could address your concerns effectively.

---

### Official Review · Reviewer_HkZx · 2024-11-10

**Soundness:** 4
**Presentation:** 4
**Contribution:** 3
**Rating:** 6
**Confidence:** 3

**Summary:**

This paper presents membership inference attacks tailored to Document Visual Question Answering (DocVQA) models, highlighting potential privacy vulnerabilities in handling sensitive document data. It introduces white-box and black-box attack methods that work without relying on auxiliary datasets, offering practical insights into real-world privacy risks. The attacks utilize features that typically arise from training data exposure. By addressing the unique complexities of multimodal data, these attacks outperform current baselines, underscoring the need for improved privacy safeguards in DocVQA systems.

**Strengths:**

- Fills a critical gap in privacy research for multimodal AI applications by designing MIA for DocVQA
- By designing attacks that operate without auxiliary datasets, the paper presents a more realistic and practical approach for assessing privacy risks in scenarios with limited data access
- The attacks can be employed in both whitebox and blackbox attacks settings
- The attacks use intuitive optimization-based discriminative features

**Weaknesses:**

- The attacks rely on repeated instances of documents in the training data, which may not always be present in real-world DocVQA applications, potentially limiting the generalizability of the approach
- While the paper identifies privacy risks, it lacks a discussion or evaluation of potential defenses that could mitigate these vulnerabilities

**Questions:**

- Given the assumption of repeated document exposure, how would the attacks perform in settings where training data consists of unique or one-time documents, with minimal repetition?
- What are some potential countermeasures or defense strategies to mitigate the privacy risks highlighted by these membership inference attacks?

---

> ### Author Response · Authors · 2024-11-21
> **Answers to Reviewer **HkZx****
>
> **W1: The attacks rely on repeated instances of documents in the training data, which may not always be present in real-world DocVQA applications, potentially limiting the generalizability of the approach**
>
> Repeated instances of documents in training data represent a realistic scenario in Document Understanding: (1) Real-world documents, such as invoices, are dense with information. Standard datasets for tasks like document key information extraction or Document VQA often feature documents with multiple questions targeting key information (e.g., PFL[1] (Section 3); TAT-QA[2] (Section 2.2); DocVQA[3] (Section 3.1); InfographicsVQA[4] (Section 3.1) etc.) and (2) State-of-the-art Document VQA models primarily process each question-answer pair independently as the default training setting. Consequently, a single document may appear multiple times in the training data, each time paired with a different question.
>
> In this work, we use two large-scale datasets, DocVQA and PFL, which include sensitive documents (e.g., invoices, industry documents). These datasets reflect scenarios where documents contain sensitive information, making non-private DocVQA models trained on such data vulnerable to membership inference. Appendix A Figure 3 in the revised paper shows the distribution of the number of questions per document for these datasets, highlighting that most documents are associated with multiple question-answer pairs.
>
> [1] Tito et al. "Privacy-Aware Document Visual Question Answering", ICDAR (2024).
>
> [2] Zhu et al. "TAT-QA: A Question Answering Benchmark on a Hybrid of Tabular and Textual Content in Finance", ACL (2021).
>
> [3] Matthew et al. "DocVQA: A Dataset for VQA on Document Images", WACV (2021).
>
> [4] Matthew et al. "InfographicVQA", arXiv:2104.12756 (2021).
>
> **Q1: Given the assumption of repeated document exposure, how would the attacks perform in settings where training data consists of unique or one-time documents, with minimal repetition?**
>
> We thank the reviewer for suggesting this defense. To evaluate the robustness of our attack strategies in this scenario, we analyze documents with minimal repetition in training to minimize the risk of memorization. Specifically, we consider a subset of member documents in $D_\text{test}​$ with only one training question and compute the membership prediction accuracy. Our results show that the attack remains effective on this subset, demonstrating its robustness under challenging conditions with minimal repetition, as shown in the table below (the size of the subset are in parenthesis). Further details on these experimental results are provided in Appendix F.4.
>
> |||PFL(1)||DVQA(51)||
> |---|:---:|:---:|:---:|:---:|:---:|
> |&nbsp;|VT5|Donut|VT5|Donut|Pix2Struct-B|
> |**FL**|0|100|86.27|88.24|90.2|
> |**IG**|100|100|90.2|56.86|88.24|
>
> In Section G of the Appendix (Defenses), we discuss another ad hoc defense that restricts the number of queries to one question per document in a black-box setting post-deployment. While these two ad hoc defenses may mitigate privacy risks, they lack the theoretical guarantees provided by Differential Privacy (DP).
>
> **W2: While the paper identifies privacy risks, it lacks a discussion or evaluation of potential defenses that could mitigate these vulnerabilities
> Q2: What are some potential countermeasures or defense strategies to mitigate the privacy risks highlighted by these membership inference attacks?**
>
> While using ad-hoc solutions, such as limiting the number of queries, might be tempting to protect against the attack, such approaches do not provide any strong theoretical guarantees. Therefore, using a more robust defense based on differential privacy is highly recommended, as it allows us to upper bound the privacy leakage by a certain privacy budget $\varepsilon$. However, DP has its own cost, as it introduces a certain amount of noise during the training process, which impacts the model's utility. To provide insights into the privacy-utility tradeoff in DP, we present the results of the attacks for different privacy budgets and also provide the respective performances of Donut trained on DocVQA, as presented in the table below:
>
> | &nbsp; | &nbsp; &nbsp; &nbsp; &nbsp; &nbsp; **$\varepsilon=8$** |  | &nbsp; &nbsp; &nbsp; &nbsp; &nbsp; &nbsp; &nbsp; **$\varepsilon=32$** | | &nbsp; &nbsp; &nbsp;  &nbsp; &nbsp; &nbsp; &nbsp;  &nbsp; &nbsp;**$\varepsilon=\infty$** |  |
> | ------ | :------: | :------: | :------: | :------: | :------: | :------: |
> | &nbsp; |  ANLS | F1 | ANLS | F1 | ANLS | F1 |
> | **FL** |  | 55.09 | | 58.84 |  | 73.81 |
> | **FLLora** | 19.16 | 54.94 | 21.81 | 58.94 | 50.12 | 73.81 |
> | **IG** |  | 56.29 |  | 59.35 |  | 73.52 |
>
> As expected, introducing DP into model training significantly reduces the attack performance at the cost of substantial utility degradation.
>
> We include these results in Section G of the Appendix, where we already discussed the aforementioned ad-hoc solution and the DP-based defense.

---

> > ### Comment · Reviewer_HkZx · 2024-11-30
> >
> > Thanks for the rebuttal responses. They address most of my concerns, I will maintain my positive score.

---

> > > ### Author Response · Authors · 2024-12-01
> > >
> > > Thank you for your thoughtful feedback and for maintaining your positive score; we are pleased that our responses have addressed your concerns.

---

### Author Response · Authors · 2024-11-21
**General Response**

We sincerely thank all reviewers for their insightful comments and valuable suggestions, and we appreciate the time spent on our review, which we found extremely helpful. We are pleased that the reviewers recognize our work as addressing a critical gap in privacy research for multimodal AI applications, specifically targeting the privacy risks in DocVQA models (**HkZx**, **hZiB**). We are grateful for the positive feedback on the effectiveness of our proposed DocMIA attacks, which perform well across both white-box and black-box settings (**HkZx**, **3nRY**, **7wUB**). Additionally, we appreciate the acknowledgment of our experimental setup, which is noted for being thorough, well-presented, and providing strong empirical evidence (**i62x**, **V2Bv**). Finally, we are happy that the reviewers find our approach to membership inference using optimization-based discriminative features novel and impactful (**i62x**, **hZiB**). In the following, we will carefully address the reviewers' questions, comments, and suggestions to further clarify and strengthen our work.

We summarize the major updates in the revised paper below. Newly added content is marked in **blue**, while removed details are shown with ~~**red strike-through**~~.
1. The analysis section, previously in Appendix E, is now moved to Appendix F.
2. We re-evaluate all methods in both White-box and Black-box settings using TPR at 1% and 3% FPR. The updated results are included in the Appendix E, along with Min-K and Min-K%++ added to the set of baselines.
3. We add an ablation study to evaluate the impact of our proposed features and compare them with other features. This is detailed in the Appendix F.1.
4. We include experimental results for our attacks against DP models, presented in the Appendix G.
5. We updated results for experiments with rephrased questions in Appendix F.2, now reported with TPR at 1% and 3% FPR, as well as under the black-box setting.

---

### Author Response · Authors · 2024-11-24
**Sincere thanks for the valuable and detailed reviews, as well as the helpful suggestions!**

We would like to thank the reviewers once again for their valuable comments and suggestions on our work. If our responses have adequately addressed your concerns, we kindly request you to consider raising the rating of our work. Of course, we are more than happy to address any additional questions or clarifications you may have.

---

### Author Response · Authors · 2024-11-27
**A gentle reminder**

Dear Reviewers,

We hope this message finds you well.

We have carefully considered your feedback in our responses. Could you kindly verify if we have adequately addressed your concerns? If so, we would greatly appreciate it if you could consider adjusting your initial score accordingly. Of course, we are more than happy to provide any further clarification if needed.

Thank you very much for your time and effort in reviewing our work!

Best regards,

Authors

---

### Meta-Review · Area_Chair_YqPx · 2024-12-05

**Metareview:**

The paper introduces membership inference attacks tailored specifically to Document Visual Question Answering (DocVQA) models, highlighting the privacy risks in this domain.
Reviewers raised concerns regarding generalizability, potential defenses, the need for additional ablation studies, and comparisons with existing defenses, among other points.
The authors provided detailed explanations and experimental results to address these issues.
Most of the reviewers stated that their concerns were well-resolved during the rebuttal phase.
In light of the overall positive reception, I recommend accepting the paper.

**Additional Comments On Reviewer Discussion:**

Reviewers raised concerns regarding generalizability, potential defenses, the need for additional ablation studies, and comparisons with existing defenses, among other points. The authors provided detailed explanations and experimental results to address these issues. I believe the revised manuscript meets the threshold for acceptance.

---

### Decision · Program_Chairs · 2025-01-22

Accept (Poster)